# Self-Referential Encoding on Modules of Anticodon Pairs—Roots of the Biological Flow System

**DOI:** 10.3390/life7020016

**Published:** 2017-04-06

**Authors:** Romeu Cardoso Guimarães

**Affiliations:** Laboratório de Biodiversidade e Evolução Molecular, Departamento de Biologia Geral, Instituto de Ciências Biológicas, Universidade Federal de Minas Gerais, Belo Horizonte, Minas Gerais 31270-901, Brasil; romeucardosoguimaraes@gmail.com or romeucg@icb.ufmg.br; Tel.: +55-31-98897-6439; Fax: +55-31-3274-4988

**Keywords:** genetic code, (proto)tRNA Dimer-Directed Protein Synthesis, self-reference, modularity, error-compensation, metabolism chronology, protein stability, punctuation

## Abstract

The proposal that the genetic code was formed on the basis of (proto)tRNA Dimer-Directed Protein Synthesis is reviewed and updated. The tRNAs paired through the anticodon loops are an indication on the process. Dimers are considered mimics of the ribosomes—structures that hold tRNAs together and facilitate the transferase reaction, and of the translation process—anticodons are at the same time codons for each other. The primitive protein synthesis system gets stabilized when the product peptides are stable and apt to bind the producers therewith establishing a self-stimulating production cycle. The chronology of amino acid encoding starts with Glycine and Serine, indicating the metabolic support of the Glycine-Serine C1-assimilation pathway, which is also consistent with evidence on origins of bioenergetics mechanisms. Since it is not possible to reach for substrates simpler than C1 and compounds in the identified pathway are apt for generating the other central metabolic routes, it is considered that protein synthesis is the beginning and center of a succession of sink-effective mechanisms that drive the formation and evolution of the metabolic flow system. Plasticity and diversification of proteins construct the cellular system following the orientation given by the flow and implementing it. Nucleic acid monomers participate in bioenergetics and the polymers are conservative memory systems for the synthesis of proteins. Protoplasmic fission is the final sink-effective mechanism, part of cell reproduction, guaranteeing that proteins don’t accumulate to saturation, which would trigger inhibition.

Living beings are metabolic flow systems that self-construct on the basis of memories and adapt/evolve on the basis of constitutive plasticity.Life is the ontogenetic and evolutionary process instantiated by living beings.

## 1. Context

Cellular structures and functions are composed by a network that may be divided into a mostly internal segment of macromolecules, the proteins and the nucleic acids, and another of micromolecules, that communicate and exchange intimately with the environment. The genetic encoding/decoding system is a key mechanism in both of these processes. It participates in the informational, the polymer sequence order-based self-referential cycle that is the cellular nucleoprotein core, the specifically endogenous and molecular identifier of living beings (Figure 1). The decoding system mediates the translation of genetic sequences—string memories—into proteins, which are the main functional working components of the cell system. The protein synthesis machinery is centered on ribosomes, while the mRNAs, tRNAs and aminoacyl-tRNA synthetases (aRS) carry the translation triplet codes, which establish the correspondences between mRNA and tRNA base triplets and the amino acids. It may be said, with some simplification, that the main function of the other—inter-genic—sequences and of the multitude of RNAs that do not code for proteins would be regulatory, in the sense of retrieving information, processing and preparing protein-coding sequences for coordinated expression in the adequate time and space contexts [1]. There are ribozyme activities in different parts of the system, its main representative being possibly the ribosomal Peptidyl Transferase Center [2,3]. It could be said that the main catalytic function in the RNP complexes would reside in enzymes, while proteins and RNAs together help and guide each other to identify targets and substrates with precision. 

The other portion of this cycle relies upon activities of the nucleic acid-binding oligomeric segments of proteins. These are involved with stabilization of the genetic molecules and the regulation of their activities through formation of nucleoprotein complexes. A useful metaphor for the self-referential aspect of the system would be of a producer-product ensemble that can only work efficiently and along the time, when the products—proteins—are able to recognize and feedback positively upon the producers—RNA, DNA. In case such informational closure would not be effective, the producers would eventually stop acting in those directions which only resulted in wasted investment, and may attempt functions in other directions that could accomplish the self-stimulating process. It is assumed that such processes occurred in the proto-biotic realm of events, involving oligomers, and led to the encodings. They are akin to autocatalysis [4], but differ from those by the composition of distinct agents that integrate among themselves to form a system. 

In the genetic encoding process, the initial recognition of the two components—tRNA and amino acid, that is mediated by the enzyme aRS—could simply involve the binding of the protein, product of the proto-, pre-(t)RNA dimer, to this, which is the producer, and the result is the formation of an ensemble that is stabilized (Figure 2B). Stabilization *per se* has a (self)stimulatory effect upon the system. The stabilized couple would evolve into a aRS-tRNA cognate set given that (1) this initial binding would at least not harm the protein synthesis activity and in case (2) the specificity would show benefits to the chemical system in which they are immersed, starting with e.g., facilitation of the flows through reduction of turbulence and of mass and energy gradients, among other effects. The two components of this cycle produce the compositional (various recognition sites in the enzyme and in the tRNA) to discrete (one triplet—the anticodon—and one amino acid) correspondences and also the self-stimulatory properties that are at the core of the living systems [5,6].

Another self-referential cycle couples with these endogenous, uniquely cellular identity structures and establishes (hetero-referential) relations with the environment. The two cycles together comprise the definition of the cell (the living being) as a metabolic flow system. The relational cycle is composed by elements of the nucleoprotein system—enzymes, ribozymes, and its products or subcomponents, such as cofactors and carriers—and is devoted to its maintenance. The scheme of the cell is fully self-referential. The ‘auto or self’ aspect relies upon memories, guaranteeing the maintenance of identity, conservation and stability. They are of two kinds: the replicative, stored in genetic strings [8] plus the systemic, dynamically conserved in positive feedback cycles that are constructed with the products [9]. Functions are of proteins and networks, highly plastic and diversified, whose restlessness and constructiveness propel and drive adaptations and evolution [10,11,12], in concert with the metabolic flow directionality. Both also contribute to the metabolic sink effectiveness.

The setting for the origins of cells inside geochemical systems (discussion in [13,14]) considers plain continuity between the universal flow where the sink is almost virtual, that is, the entropic direction given by thermodynamic degradation, which becomes real in its application to matter, where the degradative processes are energetic and material, that is, molecular. In living beings, the sink is first identified in the consumption of amino acids at the protein synthesis system, which also involves the genetic memories. A schema of the steps would be as follows. There is (a) a push to keep the metabolic flow going, driven by the gradients of concentrations of amino acids and energy, among other organics, which is relieved and at the same time reinforced by creating the (b) protein synthesis sink (pulling), and this is kept continuously active by the (c) constitutive plasticity of the components, especially the proteins, which add diversification of kinds and (d) efficiency through the development of specificity, which may be identified with the genetic code.

The sketch in Figure 1 considers the molecular bases of the cell systems, which are the constitutive grounds or bottom layers of the living—metabolic flow systems that self-construct through memories and adapt/evolve through plasticity. It is expected that this definition would be useful to accommodate other aspects of the life process—the entire set of activities (from ontogenesis to evolution) instantiated by living beings—as evolutionary additions to the basic biomolecular. The constitutive functions in the terrestrial life process are: propulsion through the plasticity properties, which is most extensive in proteins, second in RNA and less in DNA, and conservation through the replication properties of the string memory structures, especially DNA. The other propulsion subsystem is the directional metabolic flow, which is also morphogenetic and coupled concertedly with the informational subsystem. This self-referential or cyclical panorama [15] is at the basis of the model for the genetic code structure that is based on encoding directed by dimers of tRNAs (Figure 2). The bi-functional ensemble, combining propulsion/innovation/change (that are mostly in proteins) with conservation/identity/stability (mostly in replication of nucleic acids) is a basic attribute of life in general; the application of this ‘balance in mutuality’ to human affairs might consider it a source of tension and possible conflicts.

The intermingled structure of the system also means that all components are involved with the formation of a network where it is difficult to establish linear chronologies for sectors of the system. The main paradigm becomes of fully integrated coevolution [16]. When an amino acid is in the process of being encoded its availability should be guaranteed, which means that its production pathway should also be at least in the process of being fixed, as well as those routes of its utilization by the cell, and all aspects should fit each other for general shared benefits.

Some selected aspects of the model are recalled to compose this review, without much focus on technical details but attempting to encourage the reader to examine the original papers and to submit the model to further tests. It has survived already some empirical checks. We will concentrate in a few key references from which others can be reached [17].

## 2. Modular Structure in the Genetic Code and Layout of the Text

Study of the structure and formation of the genetic code has to concentrate on its elements—tRNAs with their anticodon triplets, amino acids and aRSs—that are independently moving in the cellular fluids and have to compose and aggregate for generation of function (e.g., [18,19]). We will not treat much of the work based on codons in mRNAs and genes. These are strings of triplets adjusted for translation speed, where much of the characters that were important for the encoding process do not have specific functions anymore and are not manifest in the string context. The regularities in the matrix of anticodes that indicate a structure based on dimers or pairs are here sometimes introduced in a simplified mode, through utilization of the ‘*principal dinucleotides* (pDiN) + *wobble*’ structure of the triplets, which is established by the aRSs and identify the 16 boxes; the wobble position may not be specified, becoming N. Some data are available only for dinucleotides, e.g., the hydropathies, due to low solubility of the triplets that limit the chromatographic experiments [20]. The simplification of triplets to doublets of bases is informative enough for some comparisons. The 16 boxes are organized through the formation of dimers in four modules, each with two pairs of boxes (Table 1).

The results that compose the Self-Referential Model (SRM) for the structure and the formation of the genetic code are summarized as follows. Dimers involving the triplets form network systems that are different for the self-complementary (SC) and the nonself-complementary (NSC) kinds (Table 2 shows the kinds of triplets; the networks are shown further ahead, Section 3.2). These are: SC with an R and a Y in the lateral positions of the triplet (RNY, YNR), NSC with Rs or Ys in both lateral positions (RNR, YNY). 

The networks formed by dimers of nonself-complementary anticodon triplets, after the known *en bloc* 5′A elimination [21], become asymmetric and provide an easy way to start the encoding in the modules (Section 3.2). Initiation and termination anticodes are proposed to be connected through a mechanism of competition for pairing with the initiation codons, which resulted in the deletion of the triplets corresponding to the termination codons (Section 5). 

The amino acid and protein aspects of the coding system are presented with respect to the amino acid composition of functional sequence segments or motifs (Section 4), and with respect to protein metabolic stability—half-life (Section 4.3). A synthesis of the information on both components of the system—triplets and amino acids—generated a chronology of encoding whose understanding required close examination of pathways of amino acid biosynthesis. The resulting model says that the code was formed entirely on biological-metabolic grounds, starting with one-carbon unit substrates to form amino acids, followed by gluconeogenesis (Section 6). Relevant aspects of prebiotic chemistry are discussed, facing the complexity of abiotic organics and the apparent simplicity of the C1 substrates. Models for the genetic code that are based on the sets of prebiotic abundant amino acids are considered proto-codes, possibly active in contexts where different oligomers could be adjusting their structures among themselves inside aggregates. The modules in the structure of the whole set of triplet dimers correspond to subsets of amino acids devoted to specific functions with the consequent regionalization or clustering of the functions, which would be at the basis of the property of error-reduction in the system (Section 7).

A schematic on the encoding process on the basis of proto-tRNA dimers follows. (a) In the proto-biotic realm of events the interacting compounds are oligomers. There are various options for their structures, still to be defined. (b) These can function as pre/proto tRNAs: carriers of monomers, among which there would be the amino acids. (c) They would also be apt for dimerization, through complementary sites, and the stability of the dimer structure would propitiate the transferase reaction. (d) In accordance with the types of monomers carried, they would produce of a variety of oligomers/polymers, including peptides/proteins, which is Dimer-Directed Protein Synthesis (DDPS). (e) Among the products some would be able to bind to the producers, or better, would stay bound to the producers, forming pre/protonucleoproteins. (f) When these, at the least, do not harm but maintain the producer activity and stabilize the complex, which lasts longer, the result is equivalent to stimulating the activity. (g) This is the root of a positive feedback or self-stimulating system, which is a kind of auto-catalytic network. (h) Some of these products would be structural, analogous to ribosomal proteins, others enzymatic, such as the aRSs or members of biosynthesis pathways, and so forth. (i) These compounds would manifest cohesiveness through binding protein-protein and protein-nucleic acids etc., in the route to building proto-cell globules. (j) The encoding process would require repetition of the production cycles where mutual adjustment of producers (dimers) and products (peptides) would result in specificity of binding and of catalysis, which is the beginning of fixation of the tRNA-aRS-amino acid systems. (k) It is envisaged that at some point in the process the dimers would be separated by the intromission of exogenous RNA. This would become later the messenger RNA and the process of development of individualized charging systems is enforced. 

## 3. Functionality in the Modular Organization

### 3.1. Hydropathy Correlation

The graph on the hydropathy correlation is an empirical demonstration that the set of anticodons is organized in modules [23,24], therewith providing a structure for the code system (Figure 3). The correlation is known for a long time (see [20]) but it was not conducive to modeling due to having been based on data from amino acid molecules in solution. Our reexamination utilized data on amino acid residues in proteins together with the principal dinucleotides (pDiN) of anticodons [20,23], which was fruitful in generating the self-referential model.

The rationale for the separation of modules is based on complementariness of the codes. They are seen here in a simplified way, through the pDiN (see Table 1), since data on triplets are not available due to their limited solubility. We treat further down the complexity added with consideration of the wobble base, together with other questions posed by this initial presentation of the modules, especially on the chronology of amino acid encoding and of the associated functions.

The modular structure shown in Figure 3 does not separate the two parts of the mixed sector of attributions, where the pDiN have one R and one Y base and pDiN hydropathies are intermediate. The homogeneous sector, pDiN with both bases R or both Y and hydropathies extreme, is separated into the central G:C and central A:U modules, with distinct behaviors. The central G:C should have been encoded earlier than the central A:U due to the greater thermal stability, which coincides with its simpler constitution, such as the lower number of attributions that corresponds to higher average degeneracy (Section 7.1). This rationale is extended with the correlation data: encodings start producing the non-correlated attributions (the central G:C module 1 of the homogeneous sector), then develop a moderate inclination regression line (the central A:U, module 2), and end with the steeper regression line of the mixed sector (modules 3 and 4).

Installation of the correlation in module 2 means that proteins composed by the set of ten amino acids of the homogeneous sector would be complex enough to be able to construct enzyme pockets and to develop specificity in interactions. One of such specific products is the correlation. The early aRS pockets would accommodate amino acids and proto-tRNA segments (proto-anticodons) with each other in a manner that would be more stable when they were coherent with respect to the hydropathy behavior of the partners. When both were hydrophobic they would cooperate in expelling water out of their interactive surfaces; when both were hydrophilic they would cooperate in organizing eventual water molecules in the interactive sites. When such hydropathic coherence between partners would not be established, cooperativity would fail leading to instability in the associations inside the pockets, which would hamper the fixation of a correspondence. 

The constitution of the proto-tRNA involved in the associations in the non-correlated module 1 remains an open question. The lack of correlation could be due to poorness in the amino acid constitution of the module 1 set or to a non-RNA or a pre-RNA constitution of the dimers. The lack of correlation refers to the dinucleotides but there might have been a correlation between the amino acids and whatever were the compounds in the dimers, which belong to the pre- or proto-biotic chemical systems (some non-RNA options are in [25,26]).

Exclusion of Arg from module 1 brought clearness to the interpretation based on dimers of anticodons. The hydrophilic Arg attributions traverse the sectors and are correlated coherently with the triplet kinds but do not belong to the hydroapathetic group. Difficulties in rationalization were reduced when only the Ser, Gly and Pro attributions could be interpreted as a coherent group. The ‘enigma’ of the two Ser codes, which could not be explained in any of the previous studies (see [27,28]), now receives an understanding based on dimer complementariness. The indication that the Gly and Pro codes would also have belonged to an unforeseen pair (Table 1) found support only when the interpretation provided by the metabolic pathways of amino acid biosynthesis entered the argument. Both Pro and Arg biosyntheses derive from Glu so that their encoding followed that of Glu, occurring in or after module 2. Therefore, the Pro-NGG correspondence would have been a full concession from a previous Gly-NGG, as well as Arg-YCU would have been a concession from a previous Ser-NCU that retained finally only the Ser-GCU anticode. It is indicated that the first Arg correspondence was made by the ArgRS with the YCU anticodes, in the homogeneous sector, thereafter expanding the specificity of the recognitions also to accommodate the NCG, of the mixed sector, that is, developing the 3′ U/G ambiguity. The third hexacodonic attribution shows a similar but simpler 3′R ambiguity, LeuRS for NAG and YAA.

### 3.2. Networks of Dimers in the Modules and the Encoding Process

We chose to examine the configuration of the dimer sets by accepting only the standard kind of base pair in the central position [29] and the unrestricted R:Y kinds in the lateral positions. Exclusion of base A from the 5′ position (Figure 4) is considered typical of the standard matrix of anticodes [21].

The networks distinguished, among the sets of triplets that compose the boxes (these defined by the pDiN), the self-complementary from the nonself-complementary. These kinds of triplets present different thermodynamic parameters [22] and a main differentiating property is highlighted, considered relevant for choosing among them for the protein synthesis activity of the dimers: pairs composed by molecules bearing the SC kind would be neither always nor entirely free for the hetero-referential dimerization due to the possibility of getting involved with self-dimerization [7] (Figure 2). 

A reason for the chronological precedence of the homogenous over the mixed sector of triplets would be the simplicity of the former over the complexity of the latter (Table 2). An intuitive image would take the ‘topographic landscape’ of the interaction surfaces that, in the minihelices, are in the minor grooves. The mixed sector would be presenting for interactions a non-repetitive rugged surface—bases in a string have different sizes, with bumps and depressions—in comparison with the NSC triplets of the homogeneous sector where the topology is planar, monotonically repetitive, therefore indicating a corresponding repetitiveness in the interacting partner. The rationale assumes that repetitiveness in the structures would mean simplicity both in the mechanisms for producing them and in their interactions, which would not require precise location of sites inside the repetitive segments, therewith allowing for slippages [30]. The non-repetitive structures would require some degree of specificity in the ordering or organization, and in both partners. Together with the above, there is also the symmetric structure of the NSC triplets, irrespective of the sectors, which should enlighten on the character of the primordial aRS pockets; the mature (present day) triplets are not structurally symmetric but ‘wobble + pDiN’ and the aRSs follow this organization.

Another positive attribute of the NSC dimers for the protein synthesis activity and for the encoding is consequent to the exclusion of 5′ A, which reduced the size and produced an asymmetric topology in the networks, and is repeated identically in the four modules. There are in each module two triplets with 5′ G connected to four with 5′ Y, totaling eight dimers per module. Without the exclusion of a base from the set in the 5′ position, the network would contain 16 dimers, which is the size of the SC 5′Y networks (dashed lines in blue, Figure 4A, top half). The other kind of SC dimers, with 5′G, connect between themselves through a different topology, with a total of four dimers (dashed lines in blue, Figure 4A, bottom half). Both the SC 5′G and the SC 5′Y are repeated twice and are internally fully symmetric, another component of their disadvantage in comparison with the NSC. The process of deciding on which dimer to choose for the first encoding would take longer in symmetric networks, and the degree of difficulty increases factorially with the size of the network. The encoding of the second pairs, in each of the modules, is also facilitated by the asymmetry in the network of pairs, which is a manifestation of the flow from the 1st to the 2nd members of the modules.

The suggested process of encoding (Figure 5) does not involve a typical decision process due to being almost ‘automatic’: the formation of the highest ΔG pair in the module (in the central rows of the matrix, GNG:CNC [5]) is followed by lowered concentrations of five other pairs that members of the first pair would be involved with. Two dimers are left at high concentrations, for the second encoding round, the low ΔG pairs (GNA:YNU, in the top and bottom rows of the matrix).

We could not find any interesting possibility for participation of the SC subnetworks in the encoding process. It is possible that they would be involved in integration of the networks of dimers (the NSC, diagonal, with the SC, horizontal) or in regulatory processes but this has not been specifically tested, except graphically. An integrative role is plainly acceptable for the protein associations in the Multi-aRS Complexes [31], and the aRSs in the complex would be further associated through the dimers they bring in together with them ([5,17]; an update is available as Appendix A), including the SC subnetworks.

The encoding process can be followed through studies on the aRS/tRNA interactions. We compiled data reviewed by Beuning and Forsyth [32] and displayed them according to a succession from high to low degeneracy (Table 3). The data are consistent with an evolution of the interactions from simple to complex, with respect to involvement of the anticodon bases. The high degeneracy aRSs, for the hexa- and tetracodonic amino acids, do not bind the base in the 5′ position (nucleotide 34). The aRS/tRNA recognition involves nucleotides outside the anticodon or, when they interact with the anticodon, it is done through the principal dinucleotide (nucleotides 35–36). The low degeneracy aRSs bind the three anticodon bases. There are only two exceptions to the rule, which are amino acids in the 5′G of the neighbor boxes of Tyr and His. 

### 3.3. Network Symmetry-Breaking

An important step in the encoding process was the creation of asymmetry inside the originally symmetric networks of dimers. If the encoding were to be done in the latter, some decision-type process would be required but it would take long to resolve via iterative cycles of many trials and errors with few hits, as if they were ´stagnant´, with many weak and fluctuating bindings in the network. The flow through the system would be impaired via turbulence, continuous back and forth reactions, that is undecidability. There would be many factors involved in establishing a gradient type push-pull dynamics favoring the symmetry-breaking process, not to be detailed here.

In the asymmetric network the solution comes up almost automatically, besides the expediency resulting from the reduced size (from 16 to eight dimers) of the network: the first encoding is in the pair with the highest thermal stability and the second encodings are directed to the pairs left behind at high concentration (Figure 5). Asymmetry was created by elimination of a base from the set in the 5’ position. Any base would do the job but one had to be chosen. We could not find a major component that directed the choice to A but a list of factors is offered, still to be adequately weighed and combined.
(1)Keep the (a) high stability of the G:C pair plus the (b) mono-specificity of C and bi-specificity of G. The choice becomes between A and U, where A is tri-specific and U tetra [21].(2)Avoid the very weak A:C pair at the laterals of the triplets, in favor of the G:U, which is frequent in RNA. Among the eight dimers in a module, the structure is exactly repetitive. Counting the numbers of lateral base pairs, excluding 5′A makes G:C 6, *G:U 6*, *A:C 2*, A:U 2 while excluding 5′U would make G:C 6, *G:U 2*, *A:C 6*, A:U 2.(3)Keep U due to its being the base most used for modifications, provided that these would already be present at the times of developing the 5′ elimination [33,34].(4)Keep U to profit from its tetra-fold ambiguity at decoding, as used in full in today’s vertebrate mitochondria [35].(5)Presence of A at the 5′ position in the P-site anticodon would, in a not well explained manner, destabilize the codon-anticodon pair in the A-site [36].

In view of the weakness in any of the single choices offered above, we prefer to adopt a rationale inspired on protein folding studies [37]: (a) The pressure for creating the asymmetry came via the flow. Any loss of a base would facilitate the flow from the first to the second round of encodings in a module, which was beneficial in reducing turbulence in the system caused by excess of substrates. (b) The choice for A utilized the strategy of ‘minimum harm’, which would be greater at elimination of any other base. It is rationalized that the avoidance of 5′A should be en bloc in view of the repetitive mutational formation of such kinds of tRNAs, which would be located anywhere in the space of tRNAs. This rationale is similar to that explaining the formation of eventual termination suppressors, which are continually scrutinized for deletion. Genomic data show that the absence of 5′A anticodes is at the gene level [38,39,40].

A global logic key for the encoding process produces the following summary. The homogeneous sector is encoded before the mixed sector. In each sector, the central G:C module is encoded before the central A:U module. According to the asymmetry produced by the 5′A exclusion, the high ΔG pair—whose members reside in the two central rows of the matrix, nonself-complementary GNG:CNC—is encoded first, leaving the lower ΔG pairs for the last encodings, whose members reside in the upper and lower rows of the matrix, nonself-complementary GNA:YNU. The self-complementary triplets are encoded afterwards, via expansion of the degeneracy of the first encodings in the box, which is directed to the pDiN. In case a box receives new attribution(s), triplets are conceded to them, which are usually the 5′Y anticodons to aRSs class I; exceptions are the two atypical aRSs of class II that will be treated further down.

## 4. Protein Secondary Structure, Nucleic Acid Binding and Composition of Termini

Some properties of peptides or proteins were examined with respect to amino acid composition, with the purpose of attributing functional meaning to the modules and helping in the establishment of their chronology. 

### 4.1. Secondary Structure

The homogeneous sector codes for the non-periodic protein secondary structures (Table 4; original data in [41]). All amino acids preferred in protein non-periodic segments (including coils and turns) belong to the homogeneous sector (GNPSD); those preferred in α-helices are distributed in the two sectors, namely, homogeneous (ELKR) and mixed (AMQRH); those preferred in β- strands are mostly in the mixed sector (VIYCWT), only one in the homogeneous sector (F). The same trend is built by the data on amino acids builders of Intrinsically Disordered segments of proteins (data from [42]). The latter is especially important for the model, indicating the role of non-structured segments of proteins in not constraining the kinds of structures that may be accepted for interactions and in offering open choices for development of structures after the interactions, besides the possibility of maintaining the disordered character even after the interactions [43,44,45]. 

### 4.2. RNA and DNA Binding

The homogeneous sector codes for the RNP realm (Table 5). Amino acids preferred in conserved positions of RNA-binding motifs are 75% in the homogeneous sector (GPLKFS), with only VM in the mixed sector. Amino acids belonging to the DNP realm may be preferred exclusively in DNA-binding motifs (80% in the mixed sector: AHCT; with only Glu in the homogeneous sector), or preferred in both DNA- and RNA-binding motifs (IYRQW, Arg belongs to both sectors of the code). Asp and Asn were not preferred in any nucleic acid-binding motifs. The compilation [46] does not consider highly basic motifs owing to their non-specificity for bases, interacting mostly with the sugar-phosphate backbones.

The two criteria above are linked. Radivojac et al.’s review [42] is nicely consistent with our proposition for the self-referential function of early proteins, explicitly citing the intrinsically disordered regions as characteristic of ribosomal proteins and of the splicing complex proteins. These sets encompass proteins bearing the RRM (RNA Recognition Motif, Gly-rich); the intrinsically disordered regions are also rich in Ser and Arg, which sum to the Module 1 subset (GS) of our previous list of amino acids typical of RNA-binding motifs [46]. 

### 4.3. Protein Termini and the N-End Rule

Data from the N-end rule of protein stabilization against degradation [47,48,49,50] and from statistical frequency of amino acids in the N- and C-terminal segments of proteins [51] (respectively left and right in Figure 6) are shown to superpose coherently upon each other. There is overall consistency between the two modes of examining protein strings, indicating that properties of amino acids generating the N-end rule were utilized by cells to locate stabilizing amino acids at the heads and destabilizing at the tails, which could be considered a primitive ´punctuation´ system [52].

The circularly ‘closed’ mode of presenting the code system derives from an informational description. Both the N-end rule and the protein termini regularities, plus the link between the punctuation signs (see below) conjoin to produce the informational closure drawn as a circle. Amino acids that are stabilizers of proteins against degradation are all at the left side. The drawing also requires that the codes in module 1, pairs 1 and 2, follow each other *in tandem*, initiating a central loop structure. Otherwise, amino acids of modules 3 and 4, when corresponding to central R triplets, are added backwards in the string, following from module 1 pair 1 (the initial head) to the direction of the future head (Met); the neck keeps growing until the set of amino acids is exhausted. Amino acids that are destabilizers of proteins when added to the N-ends, therefore being preferred to compose the C-ends (tails), are all at the right side of the circle. The early loop is completed by adding, still *in tandem*, module 2 to pair 2 of module 1. By the end of this loop, the amino acid Arg that fills up the homogeneous sector and initiates the mixed sector is added as the primitive tail. This is then elongated forward with addition of the amino acids in modules 3 and 4, taken from the central Y triplets, in the direction of the future tail and the termination codes. The closure is accomplished by the mechanistic and informational link between initiation and termination (topic below).

## 5. Location of the Punctuation Codes

The puzzle set forth by the indication that the boxes where the initiation Met and the termination double signs are located, respectively NAU:NUA, form a pair is only accentuated at closer examination. The real triplets do not pair due to incompatible wobble bases CAU:YUA. The other termination box, bearing the single sign UCA would not pair with the iMet but shows partial non-aligned identity, of the CA doublets. We were unable to find previous studies relating the initiation and termination codes or mechanisms. Such complication prompted a full examination of all possible connections between triplets of the related boxes: identity or complementariness, alignment or the different possible slippages, codons and anticodons. The demarcation criterion is that a mechanism should accommodate all entities involved in one same explanation: the initiation Met, alone or in combination with the second triplet, and the three X together with their neighbors in the respective boxes. The meaningful resulting combination is presented in Figure 7. The consensus constitution of the mostly virtual X anticodes, which are not present except for the suppressor instances, is YYA, minus the Trp-CCA. 

The obvious message, which nonetheless deserves highlighting is that, in all steps of the protein synthesis system, the basis is the construction of a peptide bond, which involves a couple of tRNAs. These are laterally associated in the case of translation and dimerized through an anticodon minihelix in our DDPS model for the code origin (Figure 2). Installation of initiation involved a reconfiguration of the elongation mechanisms to eliminate a wobble position between the two first codes, which may be described as a functional ‘inversion’ of the first triplet. Such inversion would be analogous to the capping process that is observed in mRNAs of eukaryotes. Our results indicate that this mechanism, at the same time, created conflicts with some anticodons that had to be eliminated therewith originating the Stop (X) codons (Figure 7). 

The connection between initiation and termination is most probably only in trans. The conflictive termination tRNAs would meet the interference sites through diffusion, not involving direct physical contacts between rostral and caudal portions of the mRNAs. Observations on circular RNA configurations pop up eventually in the bibliography but they are neither typical of nor compelling for [53] application in mRNA conformations. After the mechanism of termination tRNA exclusion was installed, some other tRNAs may mutate to acquire that constitution, then called termination or non-sense suppressors [54,55]. Such mutational events should be continually purged by purifying selection. Otherwise, the circular drawing of the code seems to be plainly justified on the basis of the informational closure character, which is also esthetically appealing to indicate that the encoding process probably reached completion in extent, while not prohibiting further evolutionary modifications inside the system. 

It could be envisaged that the punctuation system was installed upon a full set of elongation encodings. In this context, initiation would have to recode a previous triplet, which was achieved through a principal dinucleotide ‘slippage’ from the elongation ^w^CAU to the initiation CAU^w^. Another immediate consequence of the mechanism would be the creation of conflicts with competing tRNAs, whose identity, to become the terminators, would only depend on the identity of the chosen initiator. In this case, when the choice of one depends also on the possible conflictive consequences, it is suggested that the process involved some multifactorial iterative variety of trials, errors and hits. The natural case was constrained by the amino acid properties that gave rise to the N-end rule, therewith reducing the amplitude of the window of trials: initiation was directed to an N-end stabilizer, located at the N-end segment, whose conflictive tRNAs would correspond to an N-end destabilizer, located at the C-end segment. It should be interesting to investigate two related points that could test the credibility of the mechanism proposed: (a) which would be the characters of the Trp system that allowed its retention without interfering with the initiation system, and (b) is there some kind of toxicity directed to the initiation subsystem when there are nonsense suppressor tRNAs or recoded X codons [56] in the system?

### RNP World Instead of RNA World

The existence of protein translation factors that mimic the shapes of tRNAs, which include the Release Factors (RFs), would easily suggest that the latter came into the play as substitutes for the tRNAs complementary to the X codons. There are many other instances of reduction of the tRNA set, most appealing the generally forbidden 5’A anticodons. This rationale would be part of the RNA World proposition [57,58,59], indicating that RNAs started the construction of the biosystem and at some point developed the proteins as functional helpers for structures and catalysis, and developed the DNA as more stable genomic helpers, thereafter receding to the job of intermediates between those. These propositions seem to be now being made more flexible in favor of the coevolutionary panorama [16], that is, some kind of an RNP World preceding the present cellular DNP World, which is in entire consistency with the proposal of the Self-Referential Model (SRM), as sketched in Figure 2. 

The cases of protein-nucleic acid mimicry are now known to be more extensive and multifaceted [60,61,62]. The emergent consensus for the cases of tRNA mimicry in translation factors is that it refers mostly to the shapes of the tRNAs, necessary to fit the pockets and tunnels of the ribosome, while some of the functional details may differ from the expected if they would be substituting the tRNA functions [63,64]. It is realized that tRNAs are not capable of terminating protein synthesis, which requires protein activities. Termination may be achieved through a variety of ways, including the traditional absence of X anticodons, but it may occur even in the absence of the X codons, where it may rely upon directions given by other features of the mRNA 3’ end [65,66]. Even one of the paradigmatic proposed ribozyme activities, the ribosomal Peptidyl Transferase Center, has been drastically challenged [67,68] and is under reevaluation.

## 6. Metabolic Pathways

The chronology offered by the self-referential model (SRM) clashes with the traditional studies on the origins of the code [69,70,71,72,73,74,75,76,77,78]. These are centered on the set of amino acids that are produced in relative abundance under prebiotic conditions, possibly representing what would be expected from early Earth geochemical systems. Even the coevolution hypothesis, saying that much of the code organization is derived from amino acid biosynthesis pathways, starts with the geochemical set [70,71,78]. This is centered on the amino acids in the 3’C row of the matrix Val Ala Gly Asp, which could also profit from high stability of some triplet pairs such as codons Ala-GCC:GGC-Gly, less so in Val-GUC:GAC-Asp. This coincidence between an all-G:C triplet pair carrying the most abundant prebiotic amino acids gave origin to the proposals [72,76,77] based on the Ala and Gly correspondences. Note that this choice through rows has to rely upon self-complementary triplets. In the coevolution hypothesis, the hybrid molecules aminoacyl-tRNAs would be able to generate concertedly amino acid families of biosynthetic derivations and tRNA variants. The biosynthesis pathways recalled to be involved in structuring the code were the traditional heterotrophic and catabolic, centered on glycolysis, the pentose shunt and the citrate cycle. 

Our starting amino acids Gly and Ser did not fit any of the propositions above. The couple of the most abundant prebiotic is Gly and Ala. The derivation of Gly and Ser from the glycolysis route starts with a phosphorylated compound and follows the degradative path from C6 to C3 and then to C2, which is not an easy path for initial evolution to work with; the last transformations are 3-phosphoglycerate → 3-phosphohydroxypyruvate → 3-phosphoserine → Ser → Gly. A main problem with respect to the encoding process relying upon prebiotically synthesized amino acids would be the frequent and drastic fluctuations in precursor availability, approaching the ‘feast and famine’ regimes, which would not be adequate for fixation of encodings; these would require reliable and reasonably continuous sources of the substrates. In the panorama of the SRM, propositions based on the sets of prebiotic abundant amino acids could have corresponded to proto-codes where proto-tRNAs and peptides might have possibly passed through processes of mutual adjustments.

We found a simple pathway at pan-searching inside microbial biochemical diversity, the *Glycine-Serine Cycle* (Figure 8), more frequently called the *Serine Cycle* [79,80,81], which fits the SRM modular scheme. It belongs to the C1 realm of metabolism and is typical of Type II Methylotrophs (α-proteobacteria, the same group where mitochondria originated from), while having sectors of compounds that overlap other pathways and other groups of organisms. It is the simplest among central metabolic routes, starting with the C2 glyoxylate and reaching the maximum C4. It is said that nowadays its main function would be of producing AcetylCoA from C1 units. These are one CO_2_ and one reduced kind of C1, brought in by the TetraHydroFolate (H_4_F, THF) or the correlate and older H_4_MethanoPTterine (H_4_MPT) carrier. Such kind of pathway is already at the bottom row of simplicity with respect to the substrates, at the fuzzy borders between methylotrophy and autotrophy. Methylotrophs and methanotrophs are not considered autotrophs because they have wider metabolic abilities in the assimilation realm; they are able to incorporate some of the partially oxidized C1 compounds into cellular carbon before they are completely oxidized to CO_2_, this being directed to energy production. 

The Gly-Ser Cycle can be divided into (**a**) a core portion, which is the C2 + Ser cut that could be considered a nearly autonomous subcycle, and (**b**) the more complex set of the C3 derivatives from Ser plus the C4 compounds, some of which may feed directly on to the gluconeogenesis and the Citrate Cycle. Elements in the core are connected via the Gly ←→ Ser interconversion through the H_4_Folate carrier of the hydroxymethyl radical –CH_2_OH and the enzyme SHMT (Serine HydroxyMethylTransferase, SHMT, which is dependent on Pyridoxal-phosphate, PLP). This enzyme belongs to a peculiar family whose reactions and functions are much varied, sometimes called even ‘promiscuous’, which seems to fit adequately the requirements for early processes [82]. A deamination/amination reaction further unites, respectively, the Ser → HydroxyPyruvate conversion to the Glyoxylate → Glycine conversion. This core portion is nearly universally distributed and at the roots of autotrophy [83], therefore more relevant to support the earliest stage (module 1: Gly and Ser) of the genetic code. The relevance of the C3 and C4 portion for the next stage (module 2: Leu, Asp and Asn) of the code fits nicely our scheme but biochemists might question the validity of the proposal mainly in view of the supposed involvement of those reactions with oxygenated environments while it is also assumed that the code would have been formed under anoxic conditions. It is recalled that there are many possible variations on the insertion of the simple pathways of Serine synthesis, including some anaerobic [84], while the phosphorylated catabolic pathway became predominant at heterotrophy [85].

The homogeneous sector of codes contains presently other five amino acids, four of them in the Glu family plus the Phe, whose biosynthesis requires a sugar compound coming from the pentose shunt. We therefore add a stage of ‘metabolic maturation of most of the central pathways’, necessary for the completion of the homogeneous sector encodings. These pathways develop from the serine cycle C3 and C4 compounds into gluconeogenesis and then glycolysis, the pentose shunt and the citrate cycle. The key role of Glu in this stage of encodings, just at the border between the two sectors of the code, is reminiscent of the discussion asking for the causes of obligate autotrophy and obligate methanotrophy, which is supposed to reside in the sensitive step involving the loss of the α-keto glutarate dehydrogenase activity [86]. Such deficiency divides the citrate cycle into ‘halves’, the cycle assuming a ‘horseshoe’ shape: the C4 portion in one side and the C6-C5 in the other side. Such situation is also called an ‘incomplete’ citrate cycle, indicating that the cycle would have been formed or ‘completed’ through the fusion of the two sides promoted by the C5→C4 processing enzyme. It is tempting to propose that this maturation step was limiting also to the process of traversing between the code sectors via Arg^YCU^ → Arg^NCG^ since Arg is derived from Glu.

The phylometabolic approach [83] also reached the H_4_Folate/H_4_MPT roots of autotrophic pathways. The bioenergetics group of William Martin pointed to the Ljungdahl-Wood autotrophic pathway, which also has some steps involving H_4_Folate/H_4_MPT carriers but is very complex with respect to the variety of cofactors required for the biosynthesis of AcetylCoA [87,88]. A portion of this route is shared with the methanogenic pathways. Despite the simplicity of the Glycine-Serine Cycle and in face of the basal autotrophic routes for production of AcetylCoA being complex, there is the appealing possibility of biochemistry having started from geochemical C2 units, acetate and derived, which could be a reliably abundant source in some adequate environments, utilizing the H_2_ reductant produced at serpentinization [87,89,90] (I thank Yoshi Oono for this suggestion).

An apparent inconsistency in our chronology of amino acid encoding is the precocious presence of the C6 Leu, together with the C4 of module 2. Searches for proposals to substitute Leu by a simpler amino acid reach the repetitive suggestions for the central A column having been initiated with the C5 Val. Even if this suggestion is adopted, the inconsistency remains more or less of the same magnitude. The fact of Leu biosynthesis incorporating two molecules of pyruvate, the same occurring with Val biosynthesis, might make the discrepancy more acceptable in the sense of not becoming too complex with respect to sources of substrates. It might also be relevant to biochemistry the prebiotic tendency of pyruvate (the same with other α-keto acids, reactive compounds) to engage in highly complex derivations, including its dimerization into parapyruvate and formation of up to C8 compounds, that are obtained from pyruvate of carbonaceous meteorites in water [60]. Otherwise, it is possible that Ala might have been the precursor to Leu in module 2, as indicated by the existence of a variant of this kind in mitochondria of a few filamentous Saccharomycetacea and Debasyomyces [91].

### Amino Acid Hydrodynamic Size and the aRS Classes

A synthetic description of some characters of the development of encoding (Figure 9) by the aRSs can be obtained from the correlation between the increasing amino acid size [92] and biosynthetic pathway complexity (number of steps and diversity of compounds). A trend is shown, starting (module 1) with small amino acids and increasing sizes steadily to the end of the homogeneous sector, thereafter maintaining large average sizes. Variability is high in intermediate stages and lower in the sets of modules 1 and 4. Enzymes class II are typical of small and class I of large amino acids. ArgRS traverses the sectors. The Figure 9 graph can be paralleled with the hydropathy graph (Figure 3) to form groups of amino acids for the modules: the starting amino acids are (module 1) small, hydroapathetic and class II; these amino acids are among the constant and frequent oligomers of putative earliest proteins [93]. The last amino acids (module 4) are large, hydrophobic and mostly class I. Intermediate stages explore the entire hydropathy range. 

The atypical aRSs are class II for the only large amino acids of the class, and of extreme hydropathies: PheRS, acylating in the class I mode (2′), and LysRS, class I or II in different organisms, the class II being the only in the class to occupy a 5′Y set of anticodes. These occupy the last pair of triplets of the homogeneous sector GAA:YUU. The complete set of atypical aRSs includes amino acids beyond the standard 20. There are two additions, Selenocysteine (Sec, the 21st amino acid) and Pyrrolysine (Pyl, 22nd), which makes the anticodon pair $\frac{5´GAA\;PheRS}{3´UUY\;LysRS}$ only the prototype for the atypical systems. These remain all class II affecting the two Stop boxes. The class II typical acylation site is 3′ (Gly Pro Ser Ala Thr His), while Asp Asn are variable; the class I typical acylation site is 2´ (Leu Arg Glu Val Ile Met), while Cys Trp Gln Tyr are variable [94]. Distribution of the aRS classes according to the utilization of the 5′ bases in the multi-meaning boxes obtains: 5′ R is variable (class II Phe Ser^CU^ His Asp Asn/class I Ile Cys Tyr) while 5′ Y is homogeneously class I or punctuation (Leu^AA^ Ile Met Trp X Arg Gln Glu), plus the atypical Lys. There are some organisms that charge Lys via a class I enzyme, which is typical [95].

The other atypical systems share aspects with the prototypes. Charging of both amino acids Sec and Pyl repeat the LysRS class II atypical attribution to 5′Y anticodes, but in tRNAs (anticodons, UCA and CUA, respectively) that were introduced to decode Stop codons, whose tRNAs are not present (X) in the standard anticode. These cases are called recoding of X codons and differ from the termination suppression in being internal and functional in protein sequences. Termination suppression is deleterious to protein functions due to the extended anomalous C-termini. 

There is an alternative route for the charging of Cys, via the O-phosphoseryl charging of tRNA^Cys^ by a PheRS homolog (SepRS), which maintains the 2′ acylation site. The Sep-tRNA^Cys^ is thereafter transformed into Cys-tRNA^Cys^. A tRNA^Sec^ with anticodon UCA (principal dinucleotide underlined), which is not present in the standard anticode set (it corresponds to the X codon UGA), is serylated by the standard SerRS. The seryl moiety is then phosphorylated (to Sep) and this transformed to selenocysteine (Sec), to get the final product Sec-tRNA^Sec^. The pathway to Pyl starts with its synthesis from two molecules of Lys. The PylRS is homologous to PheRS but the acylation site is the 3′, typical of the class II enzymes, indicating that PylRS retains the original character of the class II that later developed the atypical function, which is shared by PheRS and SepRS [96,97].

## 7. Symmetries and Error-Reduction in Modularity

Nucleic acids should be a prime locus for the generation of symmetric structures at the acquisition of secondary or higher order organizations in view of their basic complementariness property. There have been many attempts at reaching the depths of the genetic code structure through the symmetry-search approaches (e.g., [98]) but most of them utilized codons and strings as the subject matter and are still waiting for obtaining wider support. The appeal of symmetric arrangements would reside on the possibility of facilitation of functional processes due to the repetitiveness that symmetries offer, be they derived from duplications or convergence, from direct, inverted or complementary arrangements, among other possibilities. 

Our approach is centered on the elements proper to the encoding/decoding machinery—tRNAs and their anticodons, aRSs and amino acids, considering the strings of codons later developments, either enchained triplets derived from pre-encoded tRNAs [99,100,101,102,103] or exogenous strings that acquired the ability of being translated through evolutionary adjustments. Such strings and the decoding machinery would have coevolved with focus on e.g., speed while maintaining accuracy in translation, in a process that might have not preserved the details of the process of encoding. This would be a reason for the lack of success of the studies based on the strings of codons and we hope to have uncovered some aspects of the encoding process through studies of anticodons.

### 7.1. Central Bases Compose Standard Base Pairs, Columns Are Divided into Hemi-Columns

The structure of tRNA pairs or dimers is the basal symmetric: the central base pair is of the standard Watson: Crick type while the lateral base pairs allow for the generic R:Y, keeping fixed the base in the 3’ position and choosing among the variation in the wobble position (Table 1). The primacy of the central base [29] is reflected in some regularities that are recognized in the code matrix since the time of its deciphering about half a century ago, especially the correlation of the most hydrophobic amino acids coinciding with the most hydrophobic central purine A, which is the first column.

Our approach keeps this organization, improves the correlation (Figure 3) and adds the correction that the vertical organization (in columns) derives from the juxtaposition of distinct hemicolumns. These are formed in different sets of encoding events (numbered and colored in Figure 10c), one for the homogeneous sector boxes (e.g., NAG, NAA; the central R of module 2) the other for the mixed sector boxes (e.g., NAC, NAU; the central R of module 4). The horizontal organization (in rows) is also a juxtaposition of distinct sets of encoding events. The first pairs of the four modules (1a, 2a, 3a, 4a) that start with the nonself-complementary dimers (GNG:YNC) compose the two central rows, one of them coinciding with the most traditional component of models for the code origins, which are centered on the GADV row. The self-complementary YNG:YNG and GNY:GNY of the two rows just cited are encodings secondary to the nonself-complementary. The orthogonal organization of the matrix, through columns and rows, is now the result of overlapping crisscrossed hemi-columns whose elements are pairs or dimers of triplets with a disposition that follows the diagonals inside the matrix. The full symmetry in the distribution of mono- and multi-meaning boxes (see below) is a consequence of this design. 

Symmetries are not expected to be constructed on the basis of properties of amino acids. When symmetric arrangements are observed on the basis of amino acid characters, they would probably be following the symmetry that was constructed by the triplets. In fact, we could detect three such instances (Figure 4b; one per module, except module 3, and all in the upper and lower rows) but none of them would be compelling enough to have directed research projects focusing on those arrangements. These molecules would, to the contrary, follow functional dictums of the protein segments they would be constituents of, that were examined in Section 3.1, Section 4, Section 5 and Section 6.1. 

The central base primacy in dimerization of anticodons finds a different albeit convergent counterpart in the codon-anticodon pairing. The ribosomal decoding center has an asymmetric ‘principal doublet-plus-wobble’ structure; the doublet of bases is contiguous but the third base comes from a distant segment. The contacts made by the central base pair with the ribosomal center are more numerous than those of the other bases [104]. Evidence from the aRS-tRNA interactions [34] (Table 3) indicates that the situation is highly complex. While bases are all contiguous in a triplet, there are curvatures in the anticodon loop which would be behind the development of the asymmetric interaction structure. We could attempt to suggest that the asymmetry would have derived from the process of tightly packing the adjacent tRNA anticodon loops inside the ribosome at translation of a continuous string. The forced accommodation of the loops required one of them to curve and yield space to the other. The base that was dislocated became the wobble position. Later adjustments created, among a variety of base selection and modifications, the peculiar U-turn that helps to stabilize the loop when the principal dinucleotide is only moderately stable but has a purine central base [19,105].

### 7.2. Symmetry in the Distribution of Mono- and Multi-Meaning Boxes

The picture composed by dimers in the matrix is fully symmetric and smooth (Figure 10a). The distribution of single- and multiple-meaning boxes follows the complete symmetry without any deviation (Figure 10b). These characters did not catch the attention of previous observers beyond the level of decoding where the explanation involves formation of codon: anticodon pairs [19,72,91]. The participation of anticodon dimers did not need to be invoked since questionings were satisfied by the decoding function only. 

The matrix presentation is clear enough to display the global result as the triple-circle concentric arrangement (Figure 10b), relying upon the interpretation that base composition is accompanied by the ΔG values (data in Appendix A). There is no influence of the modular structure (Figure 10c) in the final picture of the distribution of degeneracies. The inner circle is the core of the matrix, composed by single-meaning and entirely G or C principal dinucleotides. These are able to reach high thermal stability from three G:C base pairs with the correct choice of the wobble base to form nonself-complementary triplets. It is indicated that such high ∆G offered high resistance to being split into multiple attributions. The outer circle is composed by the tips of the matrix, all A or U principal dinucleotides and multiple-meaning boxes. These lower thermal stability boxes offered the least resistance to being split into multi-meanings. The principal dinucleotides in the intermediate circle have one G or C and one A or U base, therefore reaching intermediate ΔG values. A specific stabilizing factor is now recognized to be at work, namely a U-turn structure that can be formed between a central anticodonic purine base (not with a central pyrimidine base) and the U-base at position 33 [19,105]. Boxes remained with single-meanings when their triplets had a central R and developed multiple-meanings when they were central Y. The full symmetry indicates that the ΔG effects were obeyed strictly and that the affordances were profited from to saturation. Further informational necessities were fulfilled with added post-translational effects.

The reasoning above only makes sense when applied to the anticodon dimers and the aRSs that are encoding the tRNAs. Base composition is accompanied by the ΔG values and the correlation with the distribution of degeneracy per box is perfectly linear (Appendix A). The possible participation of aRSs in the dimer structures has not been studied, as well as any possible influence on thermodynamic parameters [22] between self- and nonself-complementary dimers.

### 7.3. Symmetry in Modules of Dimers and the Error-Reduction Property

The property of the encoding/decoding system of reducing the consequences of errors [106] is indicated to derive from a regionalization of the attributions or correspondences in the modules and sectors [107]. The conserved structure of the sets of triplets across modules would facilitate the encoding process from the second module onward, after the ‘learning’ at construction of module 1. The main difficulty presented to the aRS duplications would reside in adapting to different central bases and in the change from the homogeneous to the mixed sector. At the same time, each new step would benefit from the expanded repertoire of amino acids encoded in the preceding step, that is, the process incorporates cumulative and self-feeding characteristics.

Our limited list of protein properties examined supports the assertion. We detected (i) the clustering of attributions dedicated to the construction of different protein motifs (ii) inside the modules of dimers. The characters examined are coherent with the modules, which may be examined in different formats of presentation, from the graphs in Figure 3 and Figure 9 to the Table 4 and Table 5 to the circular structure of Figure 6, which may be extended into a string. The informational circle is read from the central and ancient core of elongation codes—the homogeneous sector, to both extensions, backward and forward to reach, respectively, the present configuration of the initiation and the termination sites. The circle is closed through the informational connections between initiation and termination.

The main character that oriented decisively the configuration of the circular model (Figure 6) was the N-end rule. The core of the process of building strings required tandem ligation of the first codes—module 1 (the N-ends of the cores) to the immediate followers—module 2 (the C-ends). Modules 3 and 4 are then added by staggered ligation, extending the N-ends backwards and the C-ends forwards. The attributions are shown in the order of encoding (Table 1). Functional characters of the attributions are in Table 4 and Table 5, Figure 6 and Figure 9. The strict module 1 attributions (GPS) form a conserved cluster through all characters: no hydropathy correlation, preferred in non-periodic conformations, in RNA-binding motifs and contributing to protein stabilization. At the encoding of all attributions of the homogeneous sector, amino acids preferred in nonperiodic conformations are completed and a half of the preferred in protein α-helices are added, together with some others preferred in RNA-binding motifs. Amino acids forming preferentially β-strands and DNA-binding motifs are typical of the mixed sector (modules 3 and 4). The components of the NRY quadrant of the mixed sector complete the sets of amino acids preferentially forming RNA-binding motifs and contributing to N-end stabilization. Amino acids that destabilize proteins when residing in the N-ends are all located in module 2 and in the C-end extension.

### 7.4. Metabolic Perspectives

The self-referential (SR) process, of protein products looping back to stabilize and stimulate the nucleic acids—producers, thereafter memories in RNPs—has already received contributions from diverse sources, beyond the proposition of the SRM [108,109]. Ribosomal proteins belong to the category of Intrinsically Disordered [42], which is a character of the early encodings in the SRM. Recent studies have even detected the possibility of ribosomal proteins being coded for by ribosomal RNA [102], extending and updating our older propositions [99,100] on tRNA-rRNA homologies that are consistent also with [110].

Main functional components in the self-referential processes are obviously the aRSs, whose richness and diversity of activities range much more widely than the mere role in amino acid activation and tRNA charging (e.g., [31]). It may be highly significant to the SRM propositions to find that also some of the known instances of non-ribosomal protein or peptide synthesis (NRPS) [111,112,113,114] share aspects showing similarities to what the SRM advocates. [I thank Gustavo Caetano Anollés for bringing this to my attention]. Some proteins involved in NRPS show homologies to synthetases class II, especially the SerRS, but are small and monodomain, not bearing the tRNA binding activity while preserving the amino acid activation. Other instances of proteins active in NRPS show homologies to synthetases class I, closer to the Tyr- and TrpRS, and do not activate amino acids but utilize a couple of aminoacyl-tRNAs to promote sequential reactions for peptide synthesis and cyclo-dehydration. These proteins are dated earlier than the translation machinery [110,115]. The diversity in NRPS is large but the two examples collected here suggest an intriguing analogy with the bipartite structure of tRNAs—one segment for the aminoacylation activity, the other for the anticodon. Tests are to be devised to check whether these possible ´halves´ of the synthetase activities or domains are only fortuitous results of deletions on the present-day genes or might be relics of an evolutionarily process of independent origins of the segments that were joined into the present genes. The evolutionary process of linking pieces might also leave fragile sites for genetic truncation, which will complicate the investigation. 

## 8. The Flow Is the Logic

The scheme for the development of the biological flow system draws a chain of internal sinks (Figure 1) that is initiated at the protein synthesis subsystem. All other pathways converge here, the bioenergetics plus the amino acid, nucleobase, nucleotide and nucleic acid polymer biosynthesis pathways. It reaches an end at the cytoplasmic fission process which is part of cell reproduction. The sink function keeps protein synthesis active, which drains and provides a ‘suction force’ (enzyme pockets, carrier and receptor sites empty and ‘avid’—presenting high affinity—for substrates) that shall not let the system in danger of stagnation or blockade, and of the associated toxicities. Cytoplasmic components should not accumulate to the point of saturation [116] that would inhibit protein synthesis. When some of them are difficult to get rid of, through degradation or extrusion [117,118,119,120], cytoplasmic fission is a solution. Some of the waste that cannot be degraded will aggregate into clumps that are allocated to one of the daughter cells, leaving the other healthy and clean [121].

If we consider that the picture described in the sections above answers some questions on the origins of the encoding/decoding system—the origins of biological information [122,123], the most difficult aspect to be rationalized would be on the constitution of the geochemical pre- and proto-biotic context that allowed the encoding system to be installed. After the start, the forward process—building genetic strings—can at the least be sketched with some grounding. With respect to the backward sightings, directly relevant to the triggering conditions for the initial encodings, we can only list some indications derived mostly from the hydropathy correlation and the chronology of encodings. 

Considering that the encodings required guaranteed amino acid availability from metabolic pathways, that the chronology pinpointed the pathway of assimilation of C1 units into amino acids, and assuming continuity of the prebiotic with the biologic realms, that is, the latter substituting the former and conserving some records of it, we are allowed to suggest that the steps immediately prior to the installation of the biologic process also included the production of amino acids from C1 units. Such substrates would have been abundant and relatively constant, supporting both prebiotic and biotic processes, which coincide with respect to the pathways of generating C2 compounds. It is a common theme among the abiotic organics the higher concentration of acidic compounds, which include the moderately reactive α-keto acids whose amination dampens their reactivity through the formation of amino acids. These are obtained in meteorites at concentrations in parts-per-million while nucleobases are in the range of parts-per-billion [69,124]. These became an important kind of monomeric substrate for the dimer-directed synthesis of chains, including peptide chains (DDPS). Our focus on the biomass aspect, via Gly and Ser, coincides with the bioenergetic aspect that focuses on AcetylCoA [79,83,87,88,89]. Our identification of continuity with respect to Glycine seems justified but we cannot indicate the same for Serine since prebiotic concentrations of this are generally low.

It is tempting to propose that the mechanism of dimer-directed synthesis of chains would have had a prebiotic counterpart. A reasonable prebiotic source of oligomers would be the mineral surfaces such as the crystals that compose clays [125]. Stability of the template would create repetitiveness of production which is a result equivalent to the reproduction of the oligomers. Oligomerization of slightly different monomers, such as purines and pyrimidines, or from complementary surfaces of the mother crystal layers, could create the possibility of dimerization sites via complementariness. Our hydropathy correlation data on the first encodings leaves open the question on the character of the dimerized oligomers. They might have been RNA and the lack of correlation due to the poorness of the set of encoded amino acids, but the suspicion stays that they might not have been RNA. The RNA world community has been able to offer bench modes of synthesis of nucleotides that are supposed to be possible to have happened prebiotically [126], but there is no idea of its quantification or localization, while the real observations on chondritic meteorites says of amino acids being recovered in the range of one thousand-fold more than nucleobases [69,124].

In face of the persisting doubts on the feasibility of a pure RNA world—in spite of the beauty in the RNA technology, our studies are plainly consistent with an early RNP world, late DNP. The nucleic acid and the protein component structures would have coevolved. The self-referential process depicted in Figure 2 suggests starting with the formation of RNP globules that have all components held together via protein-protein and protein-RNA binding, not requiring too much from external compartmentalization. The code is mute with respect to membrane structure or composition but it would be adequate to accept the simple proposition that the surface of globules might have been initially proteinic [127] that accommodated lipids around and in between the proteins [128], thereby generating the composite membrane structures. 

In the globules, the in versus out distinction would come, respectively, from the internal crowded macromolecular gel structures, adhered to each other, in face of the external fluid. The macromolecules develop intense motility when imbibed in water, more intensely in proteins, less in DNA. The strings are continually challenged by the polar and reactive character of water molecules, hitting either the polar exposed sites of the monomers or the bonds between the monomers of the backbones of the polymers, peptide and phosphodiester, always at the danger of hydrolysis. The trembling at the inside of the spongy structures creates continuous exchanges of materials with the environments. Those that are more necessary in the interior would be trapped there through binding. This process became more easily controllable with the help of the sequestration of some compounds by the developing lipid components.

A main component of the internal drive that keeps the flow always active is the creativity provided for by plasticity [10,11,12], which is a property most salient in proteins, less in DNA than in RNA. The starting challenge that keeps activity restless would be water and its provocation of hydrolysis which is followed by re-ligation or re-synthesis. These challenges are the sources and triggers of plasticity when they may (1) generate functional novelties, at modification of conformations and activities of the polymers and (2) generate structural and functional novelties, when re-ligation and re-synthesis produce new and different kinds of proteins and genes. Plasticity creates diversity, which are the basic richness of biosystems. The diversity is organized into structures under the influence, among many factors, of the flow itself. The flow is morphogenetic, the structures become compliant with and facilitators of the flow. I recall having heard in the early 1970s from the late David P. Bloch and read from Stuart A. Kauffman [129], respectively, that the life process is an accelerator of the universal evolutionary flow, and that all forms of organization are modes of processing energies, to which we can add many others e.g., [130,131].

## 9. Coda and Direct Tests

A proposed test for the model, as shown in Figure 2, should not be difficult to the benchwork. Since the prebiotic oligomers proposed to have started the DDPS are not known, a proxy would be the readily available small RNAs than can self-aminoacylate [132,133] or the kinds of mini-tRNAs or mini-helices that have been utilized as substitutes for the acceptor arm of tRNAs [32]. In both cases, the oligomer sets should contain segments or loops that would provide the dimerization ability and tails that would carry the amino acid. The terminal segments should be a main subject of creativity in investigating structures and contexts adequate to facilitate the transferase reaction and mimic the ribosomal, instead of the already observed terminal addition of amino acids. Evolution of the DDPS process could be investigated through utilization of different compositions of the oligomers that will dimerize and of the amino acids offered for acylation and peptide synthesis, plus the observation of the rounds of syntheses and of the activity of precursor-product binding.

### Transition to Biological Specificity

In the pre- or proto-biotic realm, the oligomers that would function as substrate carriers, which include amino acids, and would be able to dimerize, the dimers functioning as peptidyl transferase, are considered of unknown constitution. This indication derives from the unique lack of hydropathy correlation between the first set of encoded amino acids (Gly and Ser, module 1) and their correspondent RNA anticodons (Figure 3). They are called pre- or proto-tRNAs only because the present day molecules doing those jobs are the tRNAs in the ribosome. The term does not imply that they were structurally RNA-like but just says that they functionally preceded the tRNAs. The tRNAs would have entered the system in module 2, together with enzyme specificity.

Any incursion into the pre-biotic geochemical realm faces great difficulties in view of the distant time and the lack of fossil evidence. At least two kinds of hypotheses may be envisaged. The RNA World conjecture [16] presents a main problem of requiring the prebiotic existence of the highly complex nucleotides. Phosphate is not easy to obtain [134]. Sugars may be obtained from formaldehyde oligomerization, but the outcome of the reactions is a complex mixture of many kinds of compounds [135]. Nucleobases are obtained in concentrations about one thousand-fold lower than amino acids [69,124]. Composition of the nucleotide structures with the precise RNA-type links would almost certainly require enzyme catalysis, that is, they would have happened in already developed (proto)cells. There is for sure a need for starting with simpler compounds. In this case, any hypothesis that would lead to experimentation would at the same time create the additional problem of how to move from the simpler oligomer to RNA. This transition should ideally be gradual, to attempt to circumvent the problems of another transliteration or ‘translation’ system, as if reversing what happens in cells. 

In view of the difficulties in taking sides with respect to the RNA World proposition and together with the possibility of the lack of hydropathy correlation with the dinucleotides being derived only from the poorness of the encoded amino acid set—just two, probably amidst other amino acids or other compounds polymerized together with them due to catalytic non-specificity—we prefer to say of an unknown constitution of the carriers. We are not committed with the RNA World proposition and cannot offer a justified alternative. Some of the commonly accepted pre-biotic supports and guides for oligomerization reactions, e.g., from mineral surfaces such as clays [125], would be non-specific enough to accept diverse kinds of monomers. Any choice among the possibilities would have to rely upon robust experimentation.

How confident could we be on the validity of the rationale for the origins of the metabolic maze? The path from simple to complex is not to be considered ‘the’ paradigm of evolution, but it is appealing in view of having been reached here as a result, not having been taken as a premise or assumption. There is apparently no way to get simpler, in both C1 sources and in the simplicity of the C1-C4 Glycine-Serine Cycle. Methylotrophs bearing this pathway and mitochondrial ancestors trace back to the same group, the α-proteobacteria, but this is not considered among extant relatives to primitive or root organisms. Amino acids are precursors to sugars, through gluconeogenesis, and are included in pathways for biosynthesis of nucleobases. Origins of bioenergetics pathways is presently being directed to the autotrophic AcetylCoA pathway, which converges with the Glycine-Serine Cycle core. Key co-factors in these are the (H_4_)Pterins, main among these being the MethanoPterin and Folate, while the AcetylCoA pathway is much richer in cofactor requirements. In this possibly highly conflictive area, we remain. 

## Figures and Tables

**Figure 1 life-07-00016-f001:**
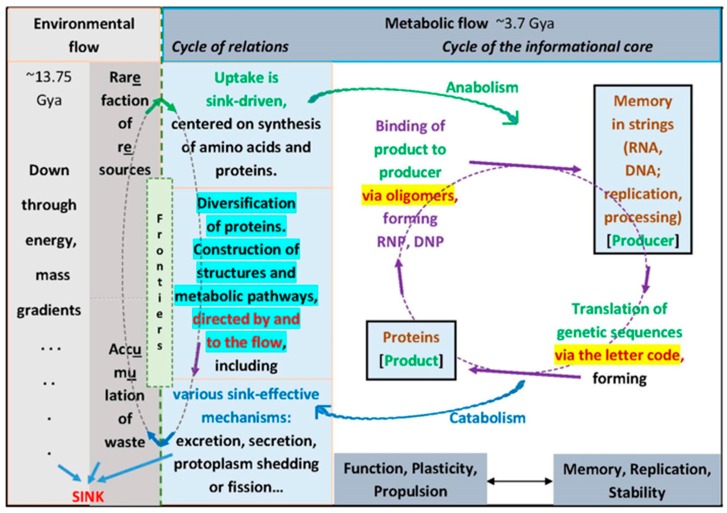
Fundamentals of living beings. Cellular functions are described as composed by two interconnected and interdependent self-referential cycles. The informational cycle is composed by the system where proteins and nucleic acids generate each other and are connected through the genetic code and the nucleic acid-binding proteins. The plasticity function, which is mainly of proteins and networks, is responsible for propulsion of the system through adaptations and evolution. The memory in strings and the replication function of nucleic acids is responsible for stability and identity that are conservation properties. The metabolism cycle is continuous with the environment that is transformed via uptake of resources and extrusion of waste. The cycles configure a flow system that is centered on the initial sink of protein synthesis. This is kept continually active through the addition of various other sink mechanisms, culminating in the extrusion of cytoplasmic chunks which gave origin to reproduction. Cycles are drawn with dashed lines indicating their limited regenerative capacity and duration, and dependence on contexts.

**Figure 2 life-07-00016-f002:**
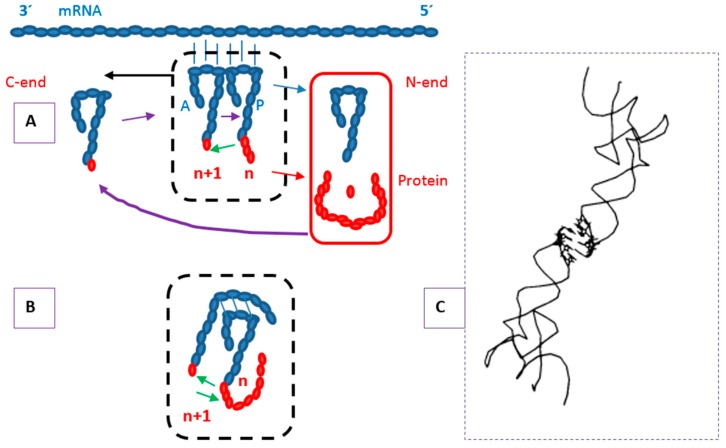
Protein synthesis directed by tRNA couples at ribosomal translation and directed by dimers of (proto)tRNAs. (**A**) Ribosomal translation of mRNA is made by couples of tRNAs whose tails, carrying an amino acid or a peptide, are placed in contact that facilitates formation of a peptide bond. Successive couples scan the mRNA from the 5′ to the 3′ termini that correspond to the protein N-end and C-ends, respectively. There are two cycles of the substrates that are fed by the aRSs. A tRNA receives an amino acid and the aminoacyl-tRNA enters the ribosomal acceptor A-site. This receives the transfer, from the P-site, of the initiator Methionine or of the nascent peptide and a peptide bond is made by the transferase activity (green arrow) at the ribosomal Peptidyl Transferase Center. The ribosome moves forward while the A-site tRNA with the n + 1 peptidyl tail is translocated to the P-site, which releases its empty tRNA that may be later re-charged. The amino acids (red elipses) derive from uptake or are recycled from protein degradation and are polymerized into proteins. These may participate in the translation system, therewith establishing a self-stimulating process (the aRSs are examples of such class of proteins) or may be released to build other structures and functions; (**B**) The self-referential model proposes that the encoding process derives from Dimer-Directed Protein Synthesis, which mimics the tRNA couples of the translation process. The anticodons are at the same time codons for each other and the dimer structure would also be a proto-ribosome, holding two tRNAs together and facilitating the transferase reaction. The tRNAs composing the dimers and the direction of the reaction could, in principle, vary along iterations of the process. Encoding would result from the binding and stabilization of the dimer structures by selected kinds of peptides produced, adequate also for keeping their functions; (**C**) A dimer of uncharged tRNAs observed experimentally [7], which survived drastic purification processes. The acceptor tails are far apart. The anticodons are held together in spite of the central base mismatch, due to the self-complementariness of the Asp anticodons $\frac{GUC}{CUG}$.

**Figure 3 life-07-00016-f003:**
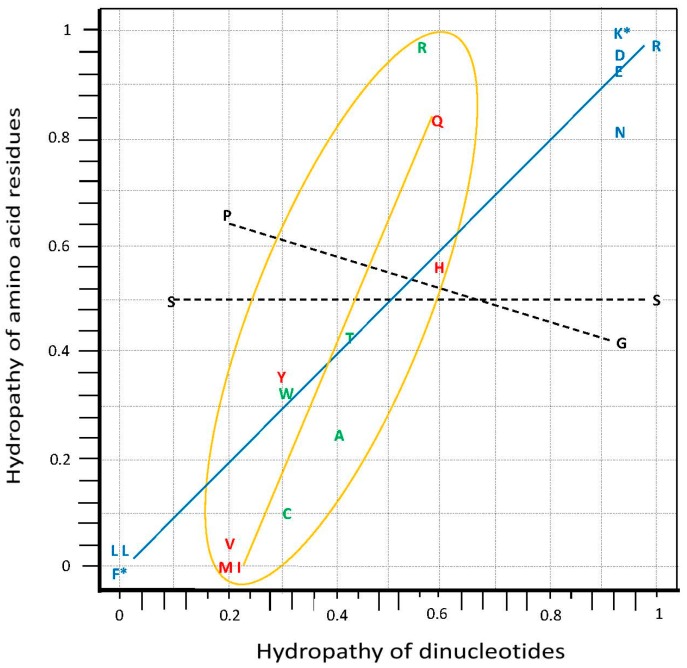
Correlation between hydropathies of amino acid residues in proteins and of dinucleotides. The dinucleotides are considered the principal dinucleotides (pDiN) of anticodons. The regularity in the distribution of anticodon types is clear [23]. The sector of mixed pDiN (circled in yellow and with a steep regression line), composed of one R and one Y, has intermediate hydropathies but the modules, one with the central bases G:C (amino acids in green), the other A:U (amino acids in red), are not resolved. The homogeneous pDiN sector (strands RR or YY) has extreme pDiN hydropathies and is separated into the two modules: hydropathies in the central G:C are not correlated (amino acids in black) but those in the central A:U are correlated (amino acids in blue). Arginine is the only amino acid belonging to both sectors. Note the extreme hydropathies of Lys and Phe; the * labels refer to the atypical character of their synthetases, commented further down.

**Figure 4 life-07-00016-f004:**
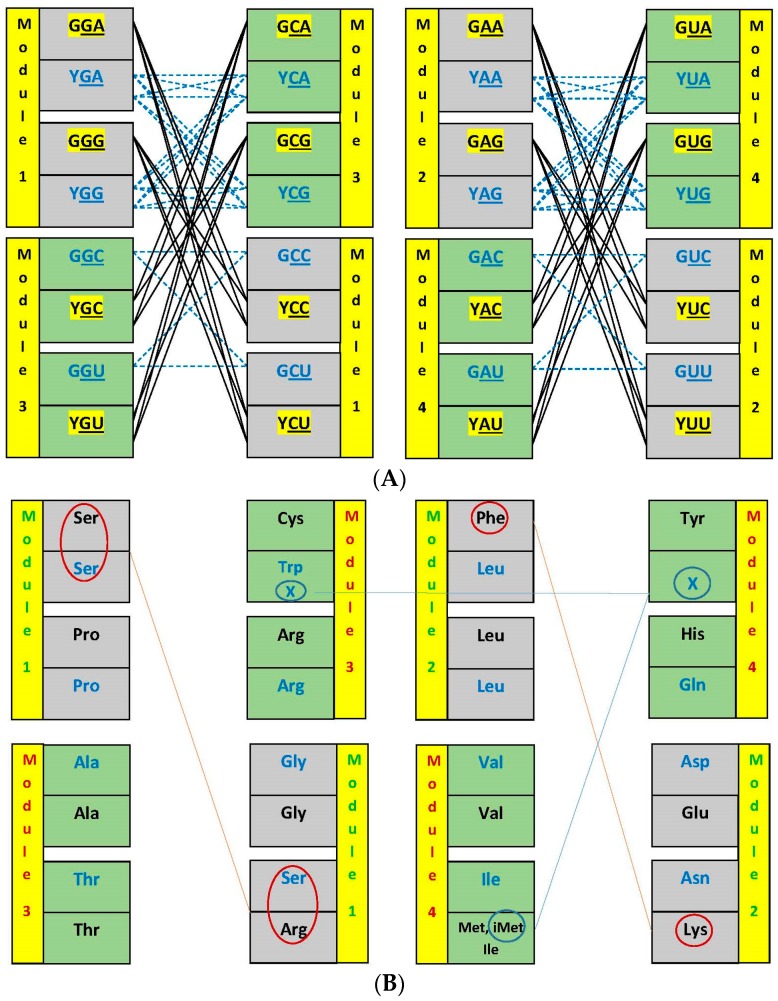
(**A**) Dimer networks. The nonself-complementary triplets (highlighted in yellow) pair among themselves forming four identical networks, combining the central G:C triplets or the central A:U triplets in each of the homogenous or mixed principal dinucleotide (pDiN) sectors. Connections in this graph follow the diagonal black lines. These networks are asymmetrical due to the exclusion of the base A from the 5′ position, which is typical of the standard anticode [21], combining two 5′G with four 5′ Y triplets (8 pairs). The self-complementary triplets (blue font) pair among themselves (connections follow the rows, blue dashed lines) forming two kinds of networks: the 5′ Y triplets are untouched by the 5′ A exclusion and the network of pairs among them is symmetric of size 16 (4 × 4); the network of the 5′G triplets is also symmetric but of size 4 (2 × 2). Chronological numbering of the modules (see text) follows both triplet kinds (homogeneous before mixed and, in each, central G:C before A:U) and amino acid characters (metabolic pathways and protein properties; see text); (**B**) Meanings of the triplets in the dimer network format. The three instances where the meanings would indicate the dimer-directed diagonal organization (in circles) are either partial (the Ser-NGA : Ser-NCU pair is corrupted by the Arg-YCU ‘invasion’; the initiation iMet-CAU would pair with the termination X-YUA, which would require some peculiar wobble-pairing between pyrimidines, but not with the X-UCA) or not obvious to the non-informed reader in the case of the Phe-GAA: Lys-YUU pair, where the link in the pair is through the respective aRSs that are both atypical (see text).

**Figure 5 life-07-00016-f005:**
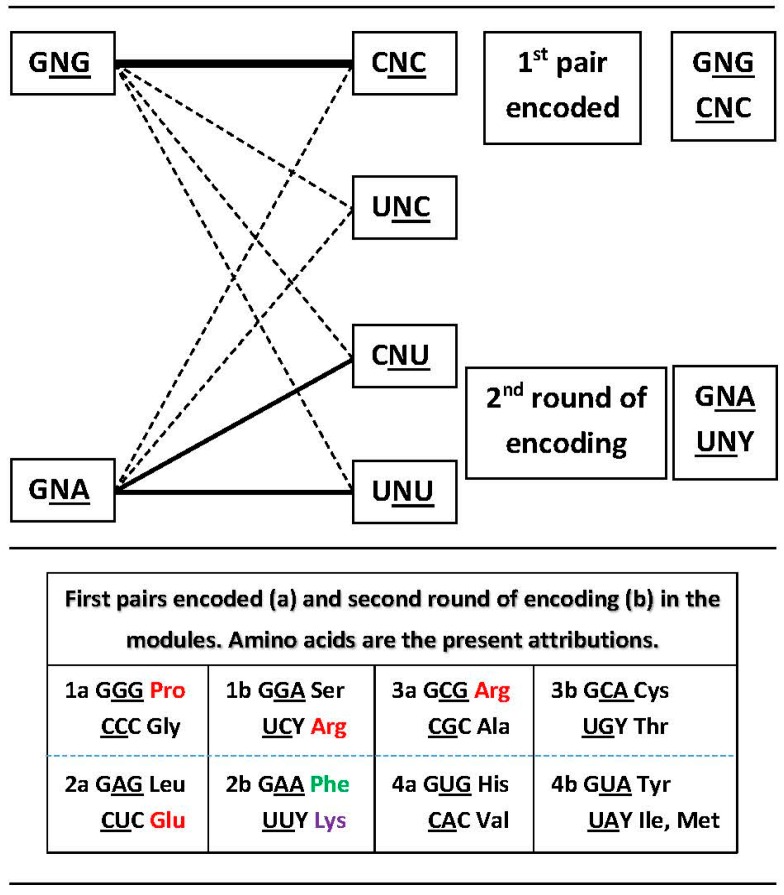
Mechanism of encoding the NSC modules. Two steps are sufficient for the description: the high ΔG pair would be stable enough to propitiate the transferase reaction and to accommodate the binding of the peptide product to itself; the five other pairs these triplets would also be involved with, of lower ΔG, become scarce, so that the two pairs left free to form are now in high concentration, which should facilitate the second round of encoding. Note that some of the present attributions in the homogeneous sector do not coincide with the constraints imposed by the metabolic pathway adopted in our scheme (see below) and require the proposition of concessions from an earlier occupier of the triplet (proposed by the metabolic scheme) to the present occupier. Three of these are related to the late entrance of the Glu family of amino acids (in red font), which does not belong to the proposed first metabolic pathway—the Glycine-Serine Cycle: in module 1 pair a, GGG-Gly conceded to Pro and in pair b YCU-Ser conceded to Arg; in module 2 pair a, CUC-Asp conceded to Glu. The other is Phe, whose biosynthesis requires a sugar (green) component. There are two pathways for the biosynthesis of Lys, involving either Asp or Glu derivations. After this stage of encodings, there are no more metabolic constraints.

**Figure 6 life-07-00016-f006:**
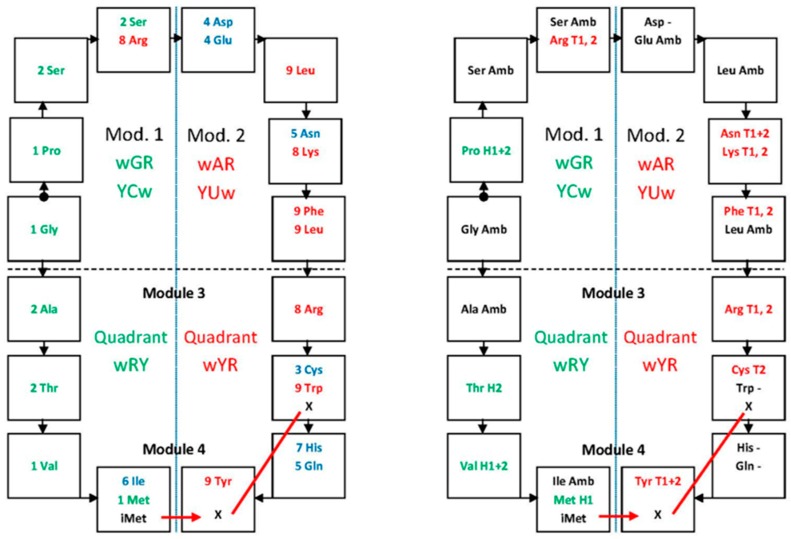
Amino acids displayed according to the N-end rule (left) and to their preferential location in protein termini (right). The N-end rule classifies amino acids as strong stabilizers of proteins against catabolism (grades 1, 2; respectively GPMV, SAT) when the half-life of proteins bearing those amino acids at the N-terminus is long; strong destabilizers (grades 8, 9; RK, LFWY) when the half-life is drastically shortened; and intermediate (grades 3–7; CDENQIH). Data on preferential location of amino acids at protein terminal segments suffer from database-dependency and from frequent ambiguities, when amino acids are preferred or avoided in both locations (Amb). H (head), significant statistical preference at the N-terminus (H1), at the second position (H2), or when the two first positions are summed as a dipeptide (H1 + 2); T (tail), significant statistical preference at the C-terminus (T1), at the second position (T2), when the preference is significant at both last positions individually (T1, 2), or when the two last positions are summed as a dipeptide (T1 + 2). In spite of the care taken by the authors with the sample sizes examined and the statistical processing, some results may suffer from deficiencies, such as the difficulties with Pro, significantly preferred in heads only in fungi, repeating the problems at trying to experimentally add it to the heads of proteins [47], and with the overall low frequency of Trp.

**Figure 7 life-07-00016-f007:**
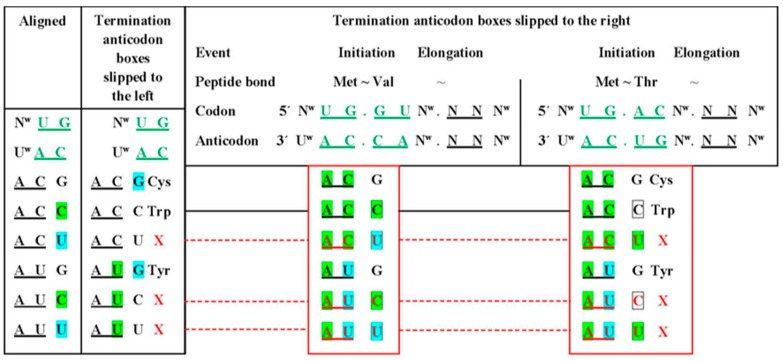
Localization of termination codes and elimination of the correspondent tRNAs directed by the initiation mechanism. **Initiation.** The principal dinucleotide of the initiation anticodon is slipped so that the wobble position is the 3’ (alternative codons would be Val-GUG and Leu YUG). The consequence is a strengthening of the initiation mechanism where there is no wobbling interposed between the two first coding triplets. The two frequent kinds of initiation couple of triplets is presented, Met-Val and Met-Thr. Note the peculiar arrangements of the contiguous principal dinucleotides, respectively, an inverted repeat wUG.GUw codons or a direct complement wUG.ACw. **Termination.** Our search for connections between the initiation and termination codes tested all possibilities of identity or complementariness and with alignment or the two kinds of slippages. Only the configuration displayed satisfied the condition of involving coherently the three termination codes: the principal dinucleotides of the boxes containing the termination codes align with the principal dinucleotide of the initiation Met and the wobble base aligns with first base of the second triplet. A conflict is instantiated where the 5’Y triplets of the cited boxes compete with the initiation Met-Val or Met-Thr anticodon doublets. This would have been the reason for eliminating the conflicting anticodons while keeping the initiator Met, which created the termination codons, void of decoding tRNAs. It is as yet not known whether the tRNA_Trp_^CCA^ was previously encoded, probably tRNA_Trp_^YCA^, therefore retained, or was encoded afterwards, being recoded from X^YCA^ which conceded X^CCA^ to Trp^CCA^; in both cases, mechanisms should be added to protect initiation from the conflict. It is highlighted the crucial determinant 3’A of the X anticodes, which conserves the Watson:Crick type pair through all tests and should have been a main guide for the X tRNA exclusions. Key: N = bases of the principal dinucleotides of the initiation triplets; N = standard complementariness of the termination anticodon base to the codon base, which is identity with the iMet anticodon; N = G:U base pair, N = termination anticodons that were deleted; N = first anticodon base whose complementariness to the first base of the second amino acid (Val or Thr) indicates the mechanism of competition with the initiation anticodon but gives no indication as to how it would have contributed to the retention or concession of the Trp anticode.

**Figure 8 life-07-00016-f008:**
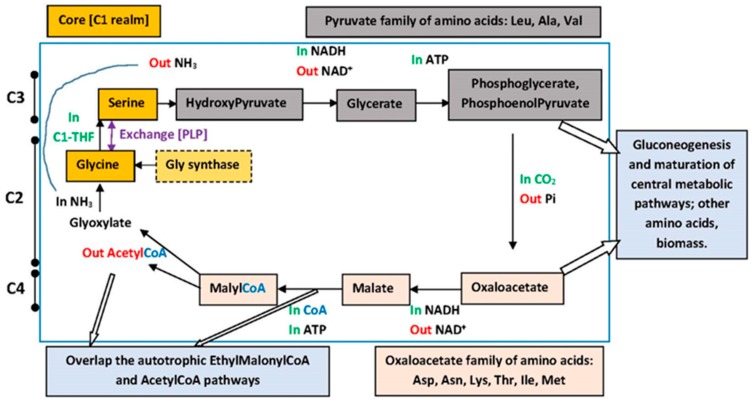
The Glycine-Serine Cycle. This is at the bottom layer of complexity among the assimilation central metabolic pathways. Just one step above the basal synthesis of Acetate, it reaches C4 compounds through additions of a reduced carbon (C1-H_4_F) and a CO_2_. Glyoxylate is aminated to Glycine and this receives a –OH-methyl to form Serine. These amino acids can feed directly the first module of the code. Various C3 derive from Ser, which can feed gluconeogenesis and synthesis of the Pyruvate family of amino acids. Addition of the CO_2_ forms the C4 compounds where the oxaloacetate family of amino acids derive, gluconeogenesis is again nourished, other key connections are established from Malate and MalylCoA. This breaks into two C2 compounds that regenerate the cycle starter and generate AcetylCoA.

**Figure 9 life-07-00016-f009:**
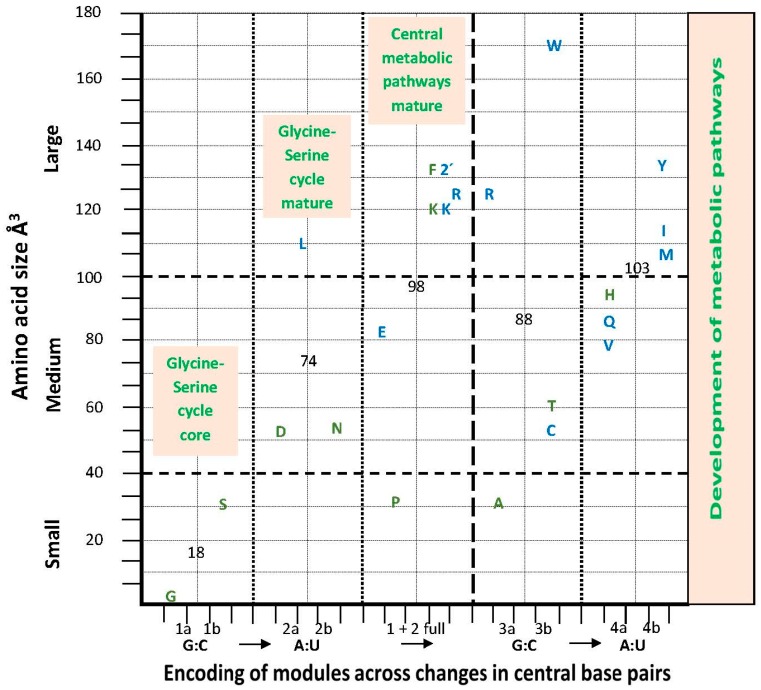
Amino acid hydrodynamic size, aRS classes and metabolic maturation. Enzymes class I in blue, class II green. Average amino acid side chain size [92] for the module in black. LysRS is uniquely class I or II (atypical) in different organisms. PheRS is also atypically a class II that acylates in the class I mode (2′). The metabolic pathways relevant to the encoding chronology are the Glycine-Serine Cycle, whose core coincides with module 1, while the C3 derivatives of Ser and the C4 compounds coincide with precursors to the amino acids in module 2. Other amino acids had to wait for development of the other central metabolism pathways, after gluconeogenesis, glycolysis, the pentose shunt and the citrate cycle. There are no metabolic constraints upon the mixed sector encodings.

**Figure 10 life-07-00016-f010:**
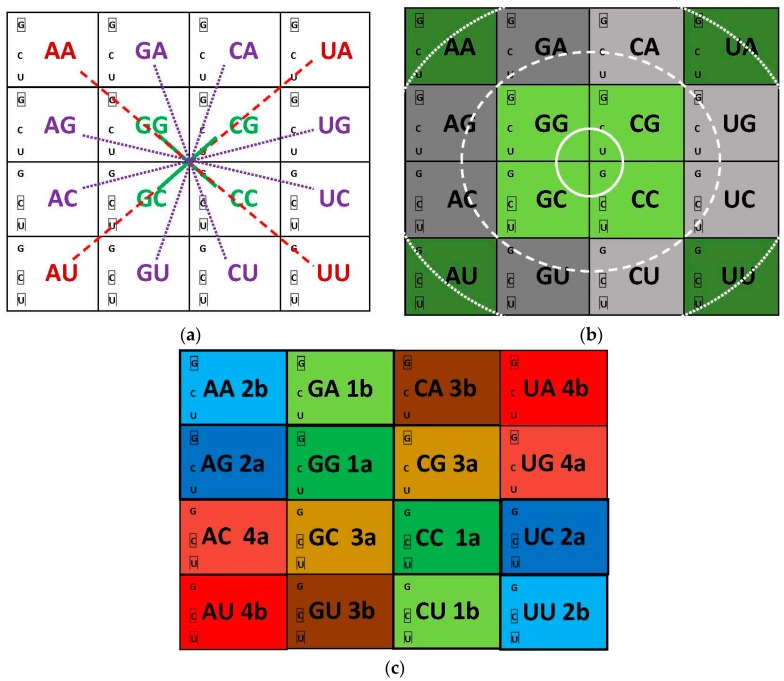
Symmetries in the organization of the matrix of anticodon dimers. They are easier to follow through the pairs of principal dinucleotides but are composed only by the nonself-complementary triplets (with the 5’ bases boxed). (**a**) Eight separate pairs are shown united by lines of different colors and shapes; (**b**) The pairs are united according to the composition of the degeneracy in the boxes: light green, inner circle, two pairs of high ∆G, principal dinucleotides with Gs or Cs only, boxes mono-meaning; dark green, outer circle, two pairs of low ∆G, principal dinucleotides with As and Us only, boxes multi-meaning; intermediate circle, principal dinucleotides with one G or C and one A or U, four pairs of intermediate ∆G, boxes mono-meaning when central base is R (dark grey), multi-meaning when central base is Y (light grey). The process ran through the organization in pairs and sectors but the final result is fully symmetric so that no hint of a deeper organization is given to the observer; (**c**) The pairs are united into modules and sectors. The format of presentation of the modules is different from the matrix in Figure 4a.

**Table 1 life-07-00016-t001:** Structure of the anticode organized through modules of dimers and the chronology of encoding. The two sectors and four modules are identified by the principal dinucleotides. The principal dinucleotides also dictate the pairing possibilities. (**A**) Bases in the 5′ position (wobble) are chosen among the variety (indicated by N), according to the generic complementariness R:Y, which is dictated by the base in the 3′ position. The central base pair is strictly of the Watson:Crick type, A:U, G:C. (**B**) Amino acids of the Glutamate family are in bold, in the homogeneous sector plus the Arg that expands into the mixed sector; Phenylalanine is the first amino acid encoded that derives from sugar precursors, also bold. Lysine biosynthesis has pathways derived from Asp or Glu. Punctuation is indicated to form after the elongation amino acid encodings; its three codes are in blue to indicate their mechanistic relatedness. Further details on the chronology will be shown later in the text.

(**A**)								
Wobble	Principal dinucleotide						→	
5’ N	Central				3’		*Anticodon*	
3’	Central				5’ N		*Anticodon*	
Principal dinucleotide					Wobble		←	
Generic R:Y	Standard G:C, A:U				Generic R:Y		*Base pair*	
(**B**)								
**Sector**		**Homogeneous Principal Dinucleotides**						
**Modules**		**Central G:C**				**Central A:U**		
**Pairs** **Elongation**		**1a**	**1b**			**2a**		**2b**
		NGG CCN	NGA UCN			NAG CUN		NAA UUN
1st occupier (5′N)		Gly Gly	Ser Ser			Leu Asp		Leu Asn
Present (5′N or 5’G/5’Y)		**Pro** Gly	Ser Ser/**Arg**			Leu Asp/**Glu**		**Phe**/Leu Asn/**Lys**
**Sector**		**Mixed Principal Dinucleotides**						
**Pairs** **Elongation**		**3a**		**3b**		**4a**		**4b**
		NCG CGN		NCA UGN		NUG CAN		NUAUAN
1st occupier (5’N)		**Arg** Ala		Cys Thr		His Val		Tyr Ile
Present (5’N or 5’G/5’Y)		**Arg** Ala		Cys/Trp Thr		His/Gln Val		Tyr Ile/Met
**Punctuation**								/Met, iMet
				/Trp, X				Tyr/X

**Table 2 life-07-00016-t002:** Splitting the boxes into 5′R and 5′Y halves and the base size ‘topographic landscape’ of the triplets. Anticodon triplets are composed of the 5′ base—the wobble position—plus the principal dinucleotide (pDiN). The 16 pDiN define the 16 boxes. The 5′ position fills boxes with options that may be undefined (N) or made explicit as G, C or Y, since the standard set contemplates the absence of 5′A [21]. Purines (R) are two-ring bulky bases and pyrimidine (Y) one-ring small bases. When they are together in a duplet or triplet, rugged topographies are presented for interactions; when the duplet or triplet is composed homogenously by one kind—all R or all Y, the landscape is smooth or planar. The self-complementary (SC) triplets may allow for self-dimerization and possibly transient internal looping since the bases in the 5′ and 3′ positions are complementary to each other [7,22]. The nonself-complementary (NSC) triplets are maintained in the extended or open configuration, and only accept hetero-dimerization—the minihelices accommodate obligatorily different kinds of triplets, obeying complementariness. The mixed pDiN is always part of rugged topography triplets; the homogeneous pDiN may lose the planar landscape configuration when it is part of a SC triplet; only the NSC triplets of the homogeneous sector maintain the full planar character. Only the NSC triplets, irrespective of sectors, maintain a symmetry center.

Sector of Principal Dinucleotide (pDiN)		Homogeneous pDiN		Mixed pDiN	
Base size configuration of the anticodon triplets		*N*RR	*N*yy	*N*Ry	*N*yR
Self-complementary triplet	Triplet	*y*RR	*G*yy	*G*Ry	*y*yR
SC	Purine (R)	••	•	••	•
lateral bases one R the other y	Pyrimidine (y)	•	••	•	••
Nonself-complementary triplet	Triplet	*G*RR	*y*yy	*y*Ry	*G*yR
NSC	Purine (R)	•••		•	• •
lateral bases both R or both y	Pyrimidine (y)		•••	• •	•

**Table 3 life-07-00016-t003:** Evolution of aRS/anticodon interactions. The data, compiled from [32] are consistent with the proposition that the initial encoding of a (proto)tRNA is dictated by the protein (aRS) binding to it, which may or may not involve the anticodon. When it involves the anticodon, it is directed to the principal dinucleotide, which means a full box degeneracy, the 5’ base being N. Evolution of proteins with the enlarged amino acid ‘alphabet’, required creation of some multi-meaning boxes, which is accompanied by interaction of the aRS with all three positions in the anticodon. The six stages are **1** = module 1; 2 = module 2; 3 = maturation of the homogeneous sector (modules 1 + 2); 4 = module 3; 5 = module 4; 6 = amino acids in the initiation NAU and the double-termination sign NUA pair of boxes.

aRS	Stage	Anticodon		Nucleotide 73	Acceptor Arm Pair			
		34	35–36		1:72	2:71	3:70	4:69
***Hexa-, Tetracodonic* (ancestral states): no interaction with the anticodon or binding through the principal dinucleotide (35–36) only**								
Ser	**1**	-		-	+	+	+	+
Leu	2	-		+	-	-	-	-
Arg	3	-	+	+Weak	-	-	-	-
Ala	4	-		+	+	+	+	-
Gly	**1**	-	+	+	+	+	+w	-
Thr	4	-	+	-	+	+	-	-
Pro	3	-	+	+	+	-	-	-
Val	5	-	+	+	-	-	-	+w
***Mono-, Dicodonic* (derived states, multi-meaning boxes): binding through the three anticodon positions**								
Gln	5	+		+	+	+	+	-
Cys	4	+		+	-	+	+	-
Met	**6**	+		+	-	+	+	-
Glu	3	+		+	+	+	-	-
Asp	2	+		+	+w	+w	-	-
Ile	**6**	+		+	-	-	-	+w
Trp	4	+		+	-	-	-	-
Asn	2	+		+	-	-	-	-
Lys	3	+		+	-	-	-	-
Phe	3	+		Weak	-	-	-	-
***Ambiguous*: dicodonic, binding in the ancestral modes**								
Tyr	**6**	-	+	+	+	-	-	-
His	5	+Weak (w)		+	+	-	-	-

**Table 4 life-07-00016-t004:** Amino acids preferred in the types of protein conformation structures.

	Sectors of Anticode Dimers (Wobble + Principal Dinucleotide)							
	Homogenous Sector, RNP Realm				Mixed Sector, DNP Realm			
**Modules of tRNA Pairs**	1 G S	2 L D N	Sector mature E P F K R	Total wRR YYw	3 R A T C W	4 V H Q I M Y	Punctu ation iM, X	Total wRY RYw
**Non-periodical, [34] Coils, Turns**	G S	D N	P	5				0
**Helices**		L	E K R	3–4	R A	H Q M		4–5
**Strands**			F	1	T C W	V I Y		6
**Disorder [35]**	S		E P K R	4–5	R	Q M		2–3
**Borderline, Neutral**	G	DN		3	A T	H		3
**Order**		L	F	2	C W	V I Y		5

**Table 5 life-07-00016-t005:** Amino acids preferred in RNA and DNA binding motifs of proteins.

	Sectors of Anticode Dimers (Wobble + Principal Dinucleotide)							
	Homogenous Sector, RNP Realm				Mixed Sector, DNP Realm			
**Modules of tRNA pairs**	1 G S	2 L D N	Sector mature E P F K R	Total wRR YYw	3 R A T C W	4 V H Q I M Y	Punctu ation iM, X	Total wRY RYw
**RNA binding [39]**	G S	L	P K F	6		V M		2
**Both**			R	0–1	R W	I Y Q		4–5
**DNA binding**			E	1	A C T	H		4

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
