# Peer review of "Self-Referential Encoding on Modules of Anticodon Pairs—Roots of the Biological Flow System"

_life, 2017, doi:10.3390/life7020016_

Round 1

Reviewer 1 Report

This review revisits the hypothesis that the genetic code arose through fixation of codon-amino acid relationship during the process of (proto-) translation. In the earliest form, the translation is suggested to occur via tRNA dimer formation where 2 tRNAs would establish an codon-anticodon relationship (each treating its base-pairing partner as an mRNA prototype). This informational self-referential system is further expanded by its relationship with (immediate) environment defined by accumulating “metabolites” (proto-metabolism) thereby establishing a form of a proto-cell.

Through the ‘principal dinucleotide’ theory, the mutual pairing of tRNA anticodons establishes the “anticode” and defines the chronology of amino acid addition to the genetic code. The first amino acids to be added are serine and glycine. But because the chronology obtained from the self-referential model disagrees with traditional studies relying on the amino acids that would be produced under prebiotic conditions, it is proposed that the early self-referential type got established through the coupling with a simple metabolic pathway, the serine-glycine cycle.

Content-wise, this manuscript is in part a review (the self-referential model) and in part a hypothesis (serine cycle coupling). Both parts tackle a very interesting subject-the origin and the stabilization of the genetic code. However, the manuscript is very hard to read, mainly because (i) the structure of the paper is inappropriate, (ii) the thoughts that are laid out lack clarity and, sometimes, logical conclusions. At times, it is hard to see why a certain section follows another: this confusion is perhaps evident when the figures or arguments are called from the succeeding (and far-away) parts of the paper. The same problem occurs even within the same section where the conclusions are sometimes laid out before the problem in question is defined.

I believe the idea of the manuscript is very interesting and it would benefit from the following:

1.     The title sections aren’t informative or are even confusing. For instance, section 2 has the title “Structure” which, in the context of the manuscript might refer to system’s structure, anticode matrix structure, geometry of the base pairing, protein folding etc. Perhaps a more definitive/descriptive titles would suit the manuscript better.

2.     In the beginning (section 1) it would be good to lay out all of the concepts discussed in the paper in a simple, clear and direct manner.

3.     Consider rearranging the order of references so that the most relevant papers appear first in the list.

4.     Consider moving some of the material into Supplementary information in order to strengthen the article’s structure (for instance, are both Table 1 and 2 needed for complete understanding of principle dinucleotide theory?).

5.     Are all references relevant and up to date: for instance, when discussing the correlation between the hydrodynamic radius of an amino acid and the distribution of aminoacyl-tRNA synthetases (aaRSs) into classes I and II a 1974 paper is cited and PheRS is mentioned as the deviation from the rule that class II tRNA synthetases utilize small amino acids. Since then, two class II aaRSs have been discovered, a pyrrolysyl-tRNA synthetase and O-phosphoseryl-tRNA synthetase; aren’t both of these also a deviation of the same sort?

6.     Figure and table legends. Some legends are incomplete or offer irrelevant information.

7.     Figure content and quality. The drawing in Figure 2 is of very poor quality; also, why include the description (and also a reference) in the figure itself?

Minor points:

tRNA synthetase rather than just synthetase.

Uniform formatting and fonts in tables.

References should be in the right format.

Author Response

Reviewer 1. Comments and Suggestions for Authors

This review revisits the hypothesis that the genetic code arose through fixation of codon-amino acid relationship during the process of (proto-) translation. In the earliest form, the translation is suggested to occur via tRNA dimer formation where 2 tRNAs would establish an codon-anticodon relationship (each treating its base-pairing partner as an mRNA prototype). This informational self-referential system is further expanded by its relationship with (immediate) environment defined by accumulating “metabolites” (proto-metabolism) thereby establishing a form of a proto-cell.

Through the ‘principal dinucleotide’ theory, the mutual pairing of tRNA anticodons establishes the “anticode” and defines the chronology of amino acid addition to the genetic code. The first amino acids to be added are serine and glycine. But because the chronology obtained from the self-referential model disagrees with traditional studies relying on the amino acids that would be produced under prebiotic conditions, it is proposed that the early self-referential type got established through the coupling with a simple metabolic pathway, the serine-glycine cycle.

Content-wise, this manuscript is in part a review (the self-referential model) and in part a hypothesis (serine cycle coupling). Both parts tackle a very interesting subject-the origin and the stabilization of the genetic code. However, the manuscript is very hard to read, mainly because (i) the structure of the paper is inappropriate, (ii) the thoughts that are laid out lack clarity and, sometimes, logical conclusions. At times, it is hard to see why a certain section follows another: this confusion is perhaps evident when the figures or arguments are called from the succeeding (and far-away) parts of the paper. The same problem occurs even within the same section where the conclusions are sometimes laid out before the problem in question is defined. Thanks. I´ll do my best to proceed the required corrections and improvements.

I believe the idea of the manuscript is very interesting and it would benefit from the following:

1.     The title sections aren’t informative or are even confusing. For instance, section 2 has the title “Structure” which, in the context of the manuscript might refer to system’s structure, anticode matrix structure, geometry of the base pairing, protein folding etc. Perhaps a more definitive/descriptive titles would suit the manuscript better. OK, I´ll do it.

2.     In the beginning (section 1) it would be good to lay out all of the concepts discussed in the paper in a simple, clear and direct manner. Thanks for the advice.

3.     Consider rearranging the order of references so that the most relevant papers appear first in the list. OK.

4.     Consider moving some of the material into Supplementary information in order to strengthen the article’s structure (for instance, are both Table 1 and 2 needed for complete understanding of principle dinucleotide theory?). Thanks for the suggestions.

The ‘principal dinucleotide’ concept – saying that code triplets have an internal structure ‘principal dinucleotide plus wobble’ – has been a point of difficult understanding by various readers, in spite of its age. This is one of the reasons why most of the work on nucleotide triplets that don’t consider this internal structure of codons or anticodons miss the point. They presume their studies are related to the encoding or decoding processes while they are only relevant to the process of enchaining triplets in strings to be translated. These strings will contain information on protein sites that are relevant for the protein functions, or on nucleic acid sites that are relevant for gene regulation or for the RNA structure, and so forth, and their study would be the same for any kind of sequences, be they extracted from protein coding genes or not. On the other side, studies devoted to understanding the genetic code have obligatorily to obey the internal structure of the codons or anticodons. This should not be considered a theory anymore, since it has received support in all aspects of the translation process. A Table in this article is constructed summarizing the data on the interaction of the aminoacyl-tRNA synthetases (aaRS) with the nucleotides at the three anticodon positions. A correspondent structure is present in the ribosomal decoding site, sometimes called the ribosomal grip that interacts simultaneously with the codon and the anticodon in a ‘triple stranded’ complex. The novelty in the present article is in identifying and distinguishing the kinds of principal dinucleotides, homogeneous and mixed, with their different attributes.

5.     Are all references relevant and up to date: for instance, when discussing the correlation between the hydrodynamic radius of an amino acid and the distribution of aminoacyl-tRNA synthetases (aaRSs) into classes I and II a 1974 (a) paper is cited and (b) PheRS is mentioned as the deviation from the rule that class II tRNA synthetases utilize small amino acids. Since then, two class II aaRSs have been discovered, a (b1) pyrrolysyl-tRNA synthetase and (b2) O-phosphoseryl-tRNA synthetase; aren’t both of these also a deviation of the same sort?

5. (a) The 1974 data were taken from the reference list in other authors’ studies on the code that are considered generally trustworthy (if I recall correctly, work of Massimo Di Giulio). I cannot criticize their choice in view of my non-biochemist background. I have never been challenged by utilizing these data and would like to receive indications of sources for revision in case they are considered necessary. The most interesting point in this scale, in comparison e. g. with molecular mass determinations, is the small hydrodynamic size of the Proline side chain that adds to the homogeneity of the module 1 group of amino acids (Gly Pro Ser).

5. (b) Thanks for pointing to the SepRS, which feeds an alternative Cys Pathway, and to the PylRS cases. The Sec pathway also belongs to these atypical novelties.

Among the total set of 20 standard amino acids plus the two additions Selenocysteine (Sec, the 21st amino acid) and Pyrrolysine (Pyl, 22nd), the anticodon pair is the prototype subset for the atypical charging systems. These are all among class II enzymes and affect the two Stop boxes. The review of Sprinzl M Chemistry of aminoacylation and peptide bond formation on the 3´ terminus of tRNA. J Biosci 311: 489-496 2006 lists the class II typical acylation site 3´ Gly Pro Ser Ala Thr His, while Asp Asn are variable; the class I typical acylation site 2´ Leu Arg Glu Val Ile Met, while Cys Trp Gln Tyr are variable. The PheRS is atypical: class II, acylating the 2´ hydroxyl. Examination of the 5´ base distribution in the multi-meaning boxes, between the aRS classes, obtains: 5´ R has variable classes (class II Phe SerCU His Asp Asn / class I Ile Cys Tyr), and 5´ Y is homogeneously class I or punctuation (LeuAA Ile Met Trp X ArgCU Gln Glu), plus the atypical Lys. There are some organisms that charge Lys via a class I enzyme, which is typical. Ambrogelly A, Söll D, Nureki O, Yokoyama S, Ibba M Class I lysyl-tRNA synthetases. NCBI Bookshelf, Madame Curie Bioscence Database [Internet] Austin TX, Landes Bioscience 2013 http://www.ncbi.nlm.nih.gov/books/NBK6444

The other atypical systems share aspects with the prototypes. Charging of both amino acids Sec and Pyl repeat the LysRS class II atypical attribution to 5´Y anticodes, but in tRNAs (anticodons, UCA and CUA, respectively) that were introduced to decode Stop codons, whose tRNAs are not present (X) in the standard anticode. These cases are called recoding of X codons and differ from the termination suppression in being internal and functional in protein sequences. Termination suppression is deleterious to protein functions due to the extended anomalous C-termini.

There is an alternative route for the charging of Cys, via the O-phosphoseryl charging of tRNACys by a PheRS homolog (SepRS), which maintains the 2´ acylation site. The Sep-tRNACys is thereafter transformed into Cys-tRNACys. Englert M, Moses S, Hohn M, Ling J, O´Donohue P, Söll D Aminoacylation of tRNA 2´- or 3´-hydroxyl by phosphoseryl- and pyrrolysyl-tRNA synthetases. FEBS Letters 587: 3360-3364, 2013

The pathway to SecRS starts with acylation of a specific tRNA[Ser]Sec with O-phosphoserine by the usual class II SerRS. The O-phosphoseryl-tRNA[Ser]Sec is then transformed into Sec- tRNA[Ser]Sec. Xu XM; Carlson BA; Mix H; Zhang Y; Saira K; Glass RS; Barry MJ; Gladyshev VN; Hatfield DL Biosynthesis of selenocysteine on its tRNA in eukaryotes. PLoS Biology 2007, 5:e4 doi 10.1371/journal.pbio.0050004

The pathway to Pyl starts with its synthesis from two molecules of Lys. The PylRS is homologous to PheRS but the acylation site is the 3´, typical of the class II enzymes, indicating that PylRS retains the original character of the class II that later developed the atypical function, which is shared by PheRS and SepRS. Englert et al 2013 

6.     Figure and table legends. Some legends are incomplete or offer irrelevant information. They will be corrected.

7.     Figure content and quality. The drawing in Figure 2 is of very poor quality; also, why include the description (and also a reference) in the figure itself? The figure was substituted by a new one.

Minor points:

tRNA synthetase rather than just synthetase. OK

Uniform formatting and fonts in tables. OK.

References should be in the right format. OK.

Reviewer 2 Report

My review of this manuscript is considerably limited by the way the manuscript is written. This probably reflects limitations in comprehension on my part but I suspect that other readers of this journal may have similar difficulties.

This manuscript is not a critical review of a particular field of study. It is a long consolidation of the author’s proposals about the roots of biology, with particular reference to the origin and evolution of the genetic code. 

Living beings are described as composed of informational and metabolic cycles. In my view metabolism and nucleic acid/protein synthesis have evolved together and are too interconnected to be separated. Life has been described as an autocatalytic system (e.g. Hordijk and Steel, 2017). What is the difference between that and ‘two interconnected and interdependent self-referential cycles’ (lines 45-46)?

Nucleic acids and nucleic acid binding proteins are proposed to interact in ‘the proto-biotic realm of events’ to form nucleoprotein complexes that ‘led to the encodings’ (lines 57-66).  Protein-nucleic acid binding stabilizes and has a self stimulatory effect on the system and evolves into a synthetase-tRNA cognate set (lines 68-70).

The central proposal/hypothesis in this manuscript rests heavily on a particular assumption, which is that an undefined ‘ensemble’ that looks a bit like a ribosomal subunit was able to bind to two entities that are somewhat like tRNAs, one of which has a 3’-polypeptide and the other a 3’-amino acid, and a peptidyl transferase-type reaction occurs producing a polypeptide with one added amino acid. The polypeptides produced are either released from the ensemble or become part of it. Therefore, one component of the ensemble is protein.  The two tRNA-like entities associate with each other through their anticodon-like triplets (Figure 2, lower left).

Even though involvement of mRNA and codon-anticodon interactions are specifically excluded in this proposal, the interactions between the anticodon triplets are assumed to have codon-anticodon type interactions in which the 5’-anticodon base of one tRNA can have wobble type variations and therefore does not have to be complementary with the 3’-base of the other tRNA. Thus, only the first two bases of an anticodon are considered relevant for the evolution of the genetic code.

The author narrows down the types of anticodon-anticodon interactions in order to explain the evolution of the genetic code. He divides anticodons into types dependent upon whether the two bases that are not in the wobble position are both purines (homogeneous) or a mix of one purine and one pyrimidine (mixed) and eliminates adenine from the wobble position because it is not involved in anticodon binding to codons. He then supposes that the former are preferable to the latter because the ‘topographical features’ are better (lines 168-170; line 296).  I don’t find this a very persuasive argument. Each type is then divided into modules.

He concludes from his analysis that Gly and Ser were the first two amino acids involved in this hypothetical type of protein synthesis. 

I have some questions about this main assumption:

1.     Are the ‘ensembles’ RNPs?

2.     If so, where do the RNA and proteins in the ensembles come from?

3.     Are the proteins randomly made of L- and D-amino acids?

4.     If only L-amino acids are encoded, how were the amino acids activated so that only L-amino acids were incorporated?

5.     Does this non-templated process produce a random set of amino acid sequences? If so, why is this advantageous?

6.     If other amino acids (e.g. Ala) were present, how did only Gly and Ser become incorporated into proteins?

7.     Are the sequences of the tRNA-like molecules random or coded? If coded, how did the coding occur?

8.     How did the peptidyl-type tRNA and deaminoacylated tRNA translocate to allow the next aminoacyl type-tRNA to bind?

9.     Why should ‘stabilization of the ensemble’ have a ‘self-stimulatory effect on the system’?  And what does this mean?

10. How did the ensemble evolve into a synthetase-tRNA set and is this set the aaRSs produced by the ribosomal process?

11. What happened to the peptidyl transferase activity when the synthetase-tRNAs evolved?

12. The crystal structure of tRNAAsp, shown in Fig 2 right side, taken from reference 4, shows binding between complementary anticodons with the sites of amino acylation widely separated.  Is there any experimental evidence for anticodon type interactions in which the 3’ ends of the tRNA-like molecules are in close proximity?

13. What is the chance that this extremely complex process would be produced randomly in a pre-biotic realm of events?

14. Why should the anticodon interactions occurring in the ensemble be similar to anticodon-codon interactions?

15. Why should A as a wobble base be excluded just because it is not found in the anticodons used in codon-anticodon interactions?

16. Do wobble bases have to be modified as they are in many cases in tRNAs to assist in anticodon-codon interactions?

Line 41. The idea that the peptidyl transferase center (PTC) is a ribozyme was suggested by early crystal structure determinations of the ribosome. However, an RNA forming a PTC does not catalyze peptide bond formation (Anderson et al., 2007), and the N-terminal tail of protein L27 is found in the PTC where it promotes peptide bond formation (Voorhees et al., 2009). The PTC is more difficult to describe as a ribozyme than it was previously.

Line 64. Oligomers were formed in the protobiotic realm of events. The author favors the hydrothermal vents scenario (line 96), but oligomers like RNA are unlikely to form in an alkaline solution inside a vent (Wächtershäuser, 2016).

Lines 96- 105. This paragraph supposes that proteins and ‘genetic memories’ (what are these memories? RNA? DNA?) are produced as a sink in a geochemical system, which is a hydrothermal vent. As far as I know, experimental evidence supporting the idea that energy arising from a difference in pH and redox state between the inside of a vent and surrounding ocean can be coupled to the production of polymers has not been demonstrated (Jackson, 2016).

Lines 106-118. This paragraph and Figure 1 are difficult for me to follow. What is a ‘constitutive bi-univocal function’? 

Following section 2 there is a lengthy discussion in sections 3-5, predicated on the main assumption, covering networks, evolution of aaRS/anticodon interactions, protein structures and functions, initiation and termination of protein synthesis.

In section 6, the author relates his conclusion that Gly and Ser were the initial amino acids to the Glycine Serine metabolic pathway, which he claims is simple, but which is clearly not simple (see Figure 8). Focusing only on the interconversion of Gly and Ser, there is an enzyme, serine transhydroxymethylase, that requires folic acid and PLP coenzymes, and a mechanism that seems to involve Tyr in proton transfer (Florio et al., 2011). It is not clear to me that a Gly/Ser protein would have had the capacity to catalyze this reaction.

In section 8, it is speculated without justification that the tRNA-type dimer-directed synthesis of proteins may have begun on mineral surfaces and ‘globules’ of RNPs were involved in formation of vesicular structures.

How this complex mechanism for protein synthesis evolved into the ribosome/mRNA/codon system for recognition of anticodons and synthesis of proteins is not addressed.

Overall, this long and difficult-to-follow manuscript is in need of significant editing. The proposal is dependent upon a central assumption about formation of ‘ensembles’ of proteins, tRNA-like entities and maybe other molecules, which have the ability to synthesize proteins using anticodon-anticodon interactions and a peptidyl transferase type reaction mechanism.  I am not a supporter of the idea that RNA was present in the prebiotic realm of events and so I find this assumption difficult to accept, but others may find it reasonable.

References

Anderson RM, Kwon M, Strobel SA. 2007 Toward ribosomal RNA activity in the absence of protein. J Mol Evol 64: 472-483.

Florio R, di Salvo M, Vivoli M, Contestabile R. 2011 Serine hydroxymethyltransferase: a model enzyme for mechanistic, structural, and evolutionary studies. Biochim. Biophys. Acta 1814:1489–1496.

Hordijk W, Steel M. 2017 Chasing the tail: The emergence of autocatalytic networks. Biosystems 152:1-10.

Jackson JB. 2016 Natural pH gradients in hydrothermal alkali vents were unlikely to have played a role in the origin of life. J Mol Evol 83:1-11.

Voorhees RM, Weixlbaumer A, Loakes D, Kelley AC, Ramakrishnan V. 2009 Insights into substrate stabilization from snapshots of the peptidyl transferase center of the intact 70S ribosome. Nat Struct Mol Biol 16:528–33.

Wächtershäuser G. 2016  In praise of error. J Mol Evol 82, 75-80.

Author Response

Reviewer 2. Comments and Suggestions for Authors

My review of this manuscript is considerably limited by the way the manuscript is written. This probably reflects limitations in comprehension on my part but I suspect that other readers of this journal may have similar difficulties.

This manuscript is not a critical review of a particular field of study. It is a long consolidation of the author’s proposals about the roots of biology, with particular reference to the origin and evolution of the genetic code. 

Living beings are described as composed of informational and metabolic cycles. In my view metabolism and nucleic acid/protein synthesis have evolved together and are too interconnected to be separated. Life has been described as an autocatalytic system (e.g. Hordijk and Steel, 2017). What is the difference between that and ‘two interconnected and interdependent self-referential cycles’ (lines 45-46)? Thanks. I add a consideration to conjoin these two approaches

Nucleic acids and nucleic acid binding proteins are proposed to interact in ‘the proto-biotic realm of events’ to form nucleoprotein complexes that ‘led to the encodings’ (lines 57-66). Protein-nucleic acid binding stabilizes and has a self stimulatory effect on the system and evolves into a synthetase-tRNA cognate set (lines 68-70).

I will rewrite these sentences to make it more clear. (a) In the proto-biotic realm of events the interacting compounds are oligomers. There are various options for their structures, still to be defined. (b) These can function as pre/proto tRNAs: carriers of monomers, among which there would be the amino acids. (c) They would also be apt for dimerization, through complementary sites, and the stability of the dimer structure would propitiate the transferase reaction. (d) In accordance with the types of monomers carried, they would produce of a variety of oligomers/polymers, including peptides/proteins. (e) Among the peptide products some would be able to bind to the producers, forming pre/protonucleoproteins. (f) When these, at the least, do not harm the producer activity and stabilize the complex, which lasts longer, the result is equivalent to stimulation of the activity. (g) This is the root of a positive feedback or self-stimulating system, which is a kind of auto-catalytic network. (h) Some of these products would be structural, analogous to ribosomal proteins, others enzymatic, such as the tRNA synthetases or members of biosynthesis pathways, and so forth. (i) These compounds would manifest cohesiveness through binding protein-protein and protein-nucleic acids etc., in the route to building proto-cell globules. (j) The encoding process would require repetition of the production cycles where mutual adjustment of producers (dimers) and products (peptides) would result in specificity of binding and of catalysis, which is the beginning of fixation of the tRNA-synthetase-amino acid systems. (k) It is envisaged that at some point in the process the dimers would be separated by the intromission of exogenous RNA. This would become later the messenger RNA and the process of development of individualized charging systems is enforced.

The central proposal/hypothesis in this manuscript rests heavily on a particular assumption, which is that an undefined ‘ensemble’ that looks a bit like a ribosomal subunit was able to bind to two entities that are somewhat like tRNAs, one of which has a 3’-polypeptide and the other a 3’-amino acid, and a peptidyl transferase-type reaction occurs producing a polypeptide with one added amino acid. The polypeptides produced are either released from the ensemble or become part of it. Therefore, one component of the ensemble is protein.  The two tRNA-like entities associate with each other through their anticodon-like triplets (Figure 2, lower left).

Even though involvement of mRNA and codon-anticodon interactions are specifically excluded in this proposal, the interactions between the anticodon triplets are assumed to have codon-anticodon type interactions in which the 5’-anticodon base of one tRNA can have wobble type variations and therefore does not have to be complementary with the 3’-base of the other tRNA. Thus, only the first two bases of an anticodon are considered relevant for the evolution of the genetic code.

The author narrows down the types of anticodon-anticodon interactions in order to explain the evolution of the genetic code. He divides anticodons into types dependent upon whether the two bases that are not in the wobble position are both purines (homogeneous) or a mix of one purine and one pyrimidine (mixed) and eliminates adenine from the wobble position because it is not involved in anticodon binding to codons. He then supposes that the former are preferable to the latter because the ‘topographical features’ are better (lines 168-170; line 296).  I don’t find this a very persuasive argument. Each type is then divided into modules.

He concludes from his analysis that Gly and Ser were the first two amino acids involved in this hypothetical type of protein synthesis. 

I have some questions about this main assumption:

1.     Are the ‘ensembles’ RNPs? Obtaining the self-referential character, which is proper of biosystems, is the aim of the model. RNPs are the final structures to be obtained. 

2.     If so, where do the RNA and proteins in the ensembles come from? RNAs and proteins would arise from mutual adjustment of the components through selection cycles. If the structure of the initial oligomers is not known, they are called pre- or proto-RNAs. A hint to the structure of the driver of the adjustment process comes from the pre-biotic abundance of glycine that is a strong component of RNA-binding motifs of proteins and the first amino acid to be encoded.   

3.     Are the proteins randomly made of L- and D-amino acids? It is obligatory to start with the mixture.

4.     If only L-amino acids are encoded, how were the amino acids activated so that only L-amino acids were incorporated? This choice derives from specificity that is indicated to arise at the encoding of the set of correspondences of module 2, which also is at the end of the sector of the homogeneous principal dinucleotides: the Gly and Ser of module 1 (dimers NGR:NCY) plus the Leu, Asp and Asn of module 2 (dimers NAR:NUY), and the whole set of 10 correspondences of the sector of correspondences (Pro, Glu, Lys, Phe, Arg; dimers NRR:NYY).

5.     Does this non-templated process produce a random set of amino acid sequences? If so, why is this advantageous? The initial amino acid set would be dominated by easiness of pre-biotic synthesis combined with the stability of the products, which corresponds approximately to the lists (also in an approximate order of decreasing abundance) compiled by Edward N Trifonov and JTF Wong (GASDEVLIPT). The early choices in our model belong to the list: module 1 (GS), module 2 (LDN), total of the homogeneous sector (add PEKFR), except for the presence of AVIT in the list that are not present in the set of homogeneous sector, here substituted by NKFR. The data in the paper show that this initial set of amino acids in the homogeneous sector is richer in those composing the RNA-binding motifs of proteins.

6.     If other amino acids (e.g. Ala) were present, how did only Gly and Ser become incorporated into proteins? The model indicates that there was a coherent development of the components of RNPs, derived from amino acids of the first two modules (e.g., Gly for purine synthesis, Asp for pyrimidine), with the biosynthesis pathways for the synthesis of those same amino acids, which was the Gly-Ser Cycle. This runs through the C3 amino acids, of which Leu is a derivation, and reaches the C4 amino acids, the oxaloacetate family, starting with Asp and Asn.  

7.     Are the sequences of the tRNA-like molecules random or coded? If coded, how did the coding occur? In the pre-biotic realm, the equivalent to replication and transcription is the simple repetitive synthesis of oligomers on mineral surfaces, such as the montmorillonite in the experiments of Jim Ferris and G Ertem. It is expected that complementariness between oligomers would arise from their synthesis being directed by the opposite surfaces that demarcate an interlayer, which would be complementary to each other. The kinds of oligomers that can be polymerized on such mineral surfaces are varied, not necessarily restricted to nucleotide components.

8.     How did the peptidyl-type tRNA and deaminoacylated tRNA translocate to allow the next aminoacyl type-tRNA to bind? The members of the dimers would be submitted to cycles of association and dissociation, possibly following environmental changes such as in temperature, pH, salinity etc. At cycling, room is made available for exchanges between the kinds of members to compose the next dimer, which may be different from the previous. Since there is no directionality in the dimer structure, it is expected that the growth of peptides would be approximately similar in size among the members of the dimers.

9.     Why should ‘stabilization of the ensemble’ have a ‘self-stimulatory effect on the system’?  And what does this mean? The terms ensemble and system refer to the same object, the first containing more of a structural connotation, the second more of the functional dynamics. Stabilization should be moderate in the sense of allowing maintenance of the functionality; if it were too strong there might be a danger of compromising the dynamics. Stability means that a certain functional state is maintained as such with longer durations – longer periods or segments of time, which should facilitate the functions. Self-stimulation would be a softer term with the same meaning of network auto-catalysis. Here we have a precursor-product system, with dimers of proto-tRNAs producing peptides that would bind the producers and stabilize their producing activity.

10. How did the ensemble evolve into a synthetase-tRNA set and is this set the aaRSs produced by the ribosomal process? The aaRS systems and the ribosomal followed different pathways while maintaining the coherence from the origins onwards. The first are enzymes with very different substrates to ligate, tRNA and amino acid. The ribosomes are very large, accommodating tRNAs inside and stabilizing them in couples whose 3’CCA tails have to be maintained close to each other for some time in order for the transferase reaction to occur, in the Peptidyl Transferase Center [a peptidyl-ribose link is broken, the tRNA is released and the peptidyl radical is incorporated into an amide bond with the A-site aminoacyl-tRNA]. 

11. What happened to the peptidyl transferase activity when the synthetase-tRNAs evolved? This process would derive from the discontinuation of the dimer-directed protein synthesis process, when exogenous RNAs were introduced, separating the members of the dimers. The exogeneous RNAs evolved into the mRNAs. The former members of dimers evolved the individual aaRS systems. Dimers would then acquire a regulatory function. The peptidyl transferase activity is now a ribosomal function, much faster and guaranteed timewise. Instead of dimer-directed, the transferase activity takes side-by-side tRNA couples.

12. The crystal structure of tRNAAsp, shown in Fig 2 right side, taken from reference 4, shows binding between complementary anticodons with the sites of amino acylation widely separated.  Is there any experimental evidence for anticodon type interactions in which the 3’ ends of the tRNA-like molecules are in close proximity? No. This is what I have been suggesting the experimentalists to test, but it seems the papers did not catch their attention yet. The dimer in the figure was not produced from any in situ functional ensemble. It was obtained from tRNAs put to dimerize and then subjected to drastic purification procedures.

13. What is the chance that this extremely complex process would be produced randomly in a pre-biotic realm of events? I follow the same expectancies that are assumed in the studies attempting to bridge the gap from geochemistry to proto-biochemistry. Reactions occurring inside surfaces of clay interlayers, inside pores of minerals, on surfaces with redox potentials such as pyrite-forming etc. Chances are difficult to estimate. The complexity involved in formation of nucleoprotein associations will obligatorily have to be tackled at some time.

One simplification that the model adds, with complexity reduced, is the possibility that the product peptide, due to its oligomer(proto-RNA)-binding property, is kept in situ, bound to the production site, so that it is not released to the solution or environment and will not have to be fished from those.

14. Why should the anticodon interactions occurring in the ensemble be similar to anticodon-codon interactions? This premise, of similarity in the two situations, is the basis of the investigation program of the Belgium group of Henri Grosjean and Dino Moras, among others, which was discontinued some time ago but gave me the support needed for the self-referential model. More recent work of H Grosjean considers, as I do also, that there was an evolutionary process of adjustment of the anticodon loop, or with the help of other ribosome-tRNA features, e. g., deriving from the introduction of nucleoside modifications, in order to homogenize (make more uniform) and reduce the differences in interaction strengths of the different kinds of triplets, to obtain more regular speed in the translation process.

15. Why should A as a wobble base be excluded just because it is not found in the anticodons used in codon-anticodon interactions? This point is very difficult to unravel and rationalize. Other authors consider that the anticodon set developed in order to decode the mRNAs. In this case, the long mRNA strings were original, and anticodons and base modifications were added to the point of reaching some satisfactory level, which did not require the usage of 5’A. I take another approach, considering that tRNAs were original, coming from small RNAs in an era where long RNAs were not present yet because the long strings would have to wait for the help of proteins. In that situation, the tRNAs were not for decoding. The process of protein synthesis was favored inside geochemical systems due to helping in establishing the flow of materials and energies, which is general to physics, biophysics and biochemistry. In this situation, encoding the tRNAs is original and translation comes later, after mRNAs (the exogenous RNAs, Question 11) entered the system.

The dimer systems are entirely symmetric in the case of presence of all four bases in the wobble position. Inside symmetric systems, interactions would go unimpeded back-and-forth with difficult flow because this requires some kind of directionality – where the flow goes to? The exclusion of one of the 5’ bases, such as the biological situation attests, would solve nicely the flow problem and we take the flow rationale as the explanation for the symmetry-breaking. At the encodings in each of the modules, the flow from the first to the second pair would be facilitated by the symmetry-breaking.

Examination of all proposals looking for some kind of difficulty that the maintenance of 5’A would introduce led to no convincing solution. We now adopt, temporarily at the least, the ad hoc proposal that the elimination was directed to the base A due to a ‘minimum harm’ effect: G and C were needed in view of their contribution to thermal stability, U was needed in view of its very wide wobbling ability, which was useful in situations of extensive simplification. Adenine was intermediate in all these attributes and could be excluded without much harm.

16. Do wobble bases have to be modified as they are in many cases in tRNAs to assist in anticodon-codon interactions? Yes, to judge from other authors’ studies.

Line 41. The idea that the peptidyl transferase center (PTC) is a ribozyme was suggested by early crystal structure determinations of the ribosome. However, an RNA forming a PTC does not catalyze peptide bond formation (Anderson et al., 2007), and the N-terminal tail of protein L27 is found in the PTC where it promotes peptide bond formation (Voorhees et al., 2009). The PTC is more difficult to describe as a ribozyme than it was previously. Thanks very much for this information. It will be added to the text. I wonder why this information did not get spread more widely until now.

Line 64. Oligomers were formed in the protobiotic realm of events. The author favors the hydrothermal vents scenario (line 96), but oligomers like RNA are unlikely to form in an alkaline solution inside a vent (Wächtershäuser, 2016). Thanks, again, for the information. My earlier reading of this subject will be changed, accordingly.

Lines 96- 105. This paragraph supposes that proteins and ‘genetic memories’ (what are these memories? RNA? DNA?) are produced as a sink in a geochemical system, which is a hydrothermal vent. As far as I know, experimental evidence supporting the idea that energy arising from a difference in pH and redox state between the inside of a vent and surrounding ocean can be coupled to the production of polymers has not been demonstrated (Jackson, 2016). The term genetic memories are for string memories, resources utilized by cells to obtain repetition of their constitutive materials, the informational biopolymers. They are usually DNA but RNA may do the same job, in some cases. Other memories are called systemic, dependent on repetitive cycles of functions that may be long lasting, such as in epigenetic systems.

I’m sorry if the idea of hydrothermal vents became so much salient in the text. I’ll redo that part. The idea for the external sink is the general flow of physics, which may be extended down to the entropic futures. In living systems, it would have started as the vast ocean basins, and so forth.

Lines 106-118. This paragraph and Figure 1 are difficult for me to follow. What is a ‘constitutive bi-univocal function’? I agree that the paragraph is difficult to follow. It will be improved. The bi-univocal indicates the ‘two-in-one’ character of the genetic system with the difficult to unravel, kind of uroboric, cyclic relationships: genes are synthesized with the help of proteins which are synthesized with the help of genes. 

Following section 2 there is a lengthy discussion in sections 3-5, predicated on the main assumption, covering networks, evolution of aaRS/anticodon interactions, protein structures and functions, initiation and termination of protein synthesis.

In section 6, the author relates his conclusion that Gly and Ser were the initial amino acids to the Glycine Serine metabolic pathway, which he claims is simple, but which is clearly not simple (see Figure 8). Focusing only on the interconversion of Gly and Ser, there is an enzyme, serine transhydroxymethylase, that requires folic acid and PLP coenzymes, and a mechanism that seems to involve Tyr in proton transfer (Florio et al., 2011). It is not clear to me that a Gly/Ser protein would have had the capacity to catalyze this reaction.

How to relate a present-day biochemical pathway to the chronology of amino acid encoding? The chronology requires a linear logic of succession, but reality would have taken many ‘revision’ measures, back and forth loops until the present state was reached. The model requires this especially for the homogeneous sector of encodings, where there is a specific pathway of biosynthesis to guide the rationale. It is not possible to attempt such measure in the mixed sector, since the metabolic maze would have reached maturity and would be too complex to allow for a criticism of the dimer-directed rationale. The precedence of the Gly-Ser core over the C3-C4 segments of the bigger cycle would seem to be well sustained, but these cover only five amino acids. The other five require the Glu precursor and a sugar component to Phe, which indicate that the central metabolic pathways reached practically full completion. The five additions over the encodings allowed by the Gly-Ser Cycle refer to four creations of multi-meaning boxes, one of them (Arg) over a box (Ser) of the module 1, and one full substitution (Gly by Pro) of the meaning of a module 1 box. Another line of justification would say that early enzymes would have been of simpler constitution, but not as much constrained as suggested in the question. We could think, for instance, of protein synthesis utilizing some initial more promiscuous aaRS or with more relaxed specificities, that would have been at work while the real mono-specificity would be established at later states of the system. The family of enzymes to which the SHMT belongs would be a good candidate for the work of developing a promiscuous and wide ‘specificity’ kind, according to the article offered by the reviewer.

In section 8, it is speculated without justification that the tRNA-type dimer-directed synthesis of proteins may have begun on mineral surfaces and ‘globules’ of RNPs were involved in formation of vesicular structures.

I’ll proceed with the corrections.

How this complex mechanism for protein synthesis evolved into the ribosome/mRNA/codon system for recognition of anticodons and synthesis of proteins is not addressed.

I tried to cover this question, in part, in the answers to Questions 11 and 15.

Overall, this long and difficult-to-follow manuscript is in need of significant editing. The proposal is dependent upon a central assumption about formation of ‘ensembles’ of proteins, tRNA-like entities and maybe other molecules, which have the ability to synthesize proteins using anticodon-anticodon interactions and a peptidyl transferase type reaction mechanism. I am not a supporter of the idea that RNA was present in the prebiotic realm of events and so I find this assumption difficult to accept, but others may find it reasonable.

I am also not a supporter of the RNA World proposition, but I did not want to buy a dispute with its supporters. The position our model supports is simply RNP before DNP. Please note that I kept the subject of replication under a low profile and brought genes to the role of memories instead of instructions. Thanks for the very attentive review. Your suggested references were all added to the text.

Reviewer 3 Report

I suggest to accept this manuscript in the present form.

Author Response

Thank you very much for your attention. 

Round 2

Reviewer 1 Report

The manuscript could be clearer. Again, it would be good to consider some of the section titles as titles such as “Punctuation” or “Protein physiology” might be too misleading in the context of the paper.

The newly introduced paragraph about selenocysteinyl-tRNA synthesis needs to be corrected because: a) there is no "SecRS", tRNASec is serylated by canonical SerRS in the first step,b) tRNASec is not phosphoserylated by SerRS, but is serylated by it and only then can be phosphorylated giving rise to Sep-tRNASer (catalyzed by PSTK) c) Sep-tRNASec is then substrate for SepSecS which converts phosphoseryl moiety into Sec.

Author Response

I am asking MDPI to proceed the text and language corrections and editing, which I´ll pay for

The section titles were extended, according to your suggestion

Correction done, see lines 764-7

Thank you very much for your attentive examination and collaboration in the improvement of the article. I have learned from our conversations.  

Reviewer 2 Report

The author has made some improvements to the presentation of his ideas but his proposal is still for me difficult to read.  I do not find persuasive the answers to the questions that I asked. I think that the central assumption that RNA was present in some form or another during the prebiotic realm of events is implausible. RNA is simply too complicated. In my opinion RNA is better regarded as the product of an extensive intracellular evolutionary process that allowed the complex pathway for RNA synthesis to develop. Having said that, many other people agree with the author on this question, and may find his proposal more reasonable and understandable than I do.     

Author Response

I am asking MDPI to proceed the text and language corrections and editing, which I´ll pay for

I added a topic 9.1 lines 1003-31 plus two references to make clearer that I don´t support the RNA World proposition 

Thank you very much for your attentive examination of the article. I have learned much from our conversation in the process. Your collaboration in the improvement of the text was of help.

Round 3

Reviewer 1 Report

I believe the revised version of the manuscript is suitable for publication.